# Large Language Models as Urban Residents: An LLM Agent Framework for Personal Mobility Generation

Jiawei Wang[1]        Renhe Jiang[1]*        Chuang Yang[1]        Zengqing Wu[2]

Makoto Onizuka[2]        Ryosuke Shibasaki[1]        Noboru Koshizuka[1]        Chuan Xiao[2]

[1]The University of Tokyo, [2]Osaka University
{jiawei@g.ecc, koshizuka@iii}.u-tokyo.ac.jp
{jiangrh,chuang.yang,shiba}@csis.u-tokyo.ac.jp
wuzengqing@outlook.com, {onizuka,chuanx}@ist.osaka-u.ac.jp

## Abstract

This paper introduces a novel approach using Large Language Models (LLMs) integrated into an agent framework for flexible and effective personal mobility generation. LLMs overcome the limitations of previous models by effectively processing semantic data and offering versatility in modeling various tasks. Our approach addresses three research questions: aligning LLMs with real-world urban mobility data, developing reliable activity generation strategies, and exploring LLM applications in urban mobility. The key technical contribution is a novel LLM agent framework that accounts for individual activity patterns and motivations, including a self-consistency approach to align LLMs with real-world activity data and a retrieval-augmented strategy for interpretable activity generation. We evaluate our LLM agent framework and compare it with state-of-the-art personal mobility generation approaches, demonstrating the effectiveness of our approach and its potential applications in urban mobility. Overall, this study marks the pioneering work of designing an LLM agent framework for activity generation based on real-world human activity data, offering a promising tool for urban mobility analysis.

Source codes are available at `https://github.com/Wangjw6/LLMob/` .

## 1   Introduction

The prevalence of large language models (LLMs) has facilitated a variety of applications extending beyond the domain of NLP. Notably, LLMs have gained widespread usage in furthering our understanding of humans and society in a multitude of disciplines, such as economy [1] and political science [2], and have been employed as agents in various social science studies [36, 35, 10]. In this paper, we target the utilization of LLM agents for the study of personal mobility data. Modeling personal mobility opens up numerous opportunities for building a sustainable community, including proactive traffic management and the design of comprehensive urban development strategies [4, 3, 45]. In particular, generating reliable activity trajectories has become a promising and effective way to exploit individual activity data [13, 6]. On one hand, learning to generate activity trajectory leads to a thorough understanding of activity patterns, enabling the flexible simulation of urban mobility. On the other hand, while individual activity trajectory data is abundant thanks to advances in telecommunications, its practical use is often limited due to privacy concerns. In this sense, generated data can provide a viable alternative that offers a balance between utility and privacy.

---

*Corresponding author.

38th Conference on Neural Information Processing Systems (NeurIPS 2024).

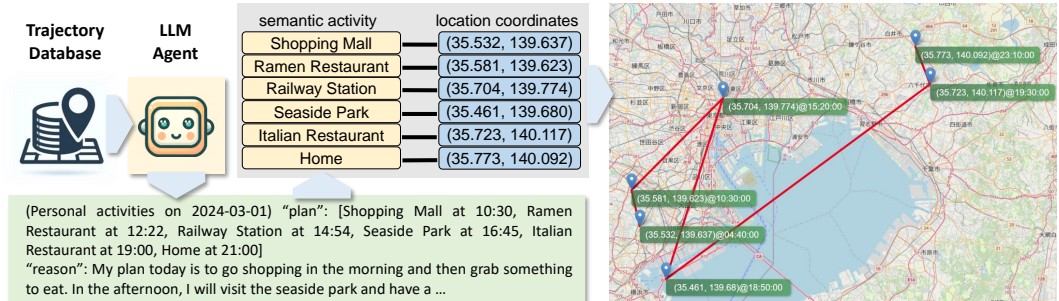

Figure 1: Personal mobility generation with an LLM agent.

While advanced data-driven learning-based methods offer various solutions to generate synthetic individual trajectories [13, 39, 9, 6, 16, 38], the generated data only imitates real-world data from the data distribution perspective rather than semantics, rendering them less effective in simulating or interpreting activities in novel or unforeseen scenarios with a significantly different distribution (e.g., a pandemic). Thus, in this study, to explore a more intelligent and effective activity generation, we propose to establish a trajectory generation framework by exploiting the emerging intelligence of LLM agents, as illustrated in Figure 1. LLMs present two significant advantages over previous models when applied to activity trajectory generation:

- **Semantic Interpretability.** Unlike previous models, which have predominantly depended on structured data (e.g., GPS coordinates-based trajectory data) for both calibration and simulation [17, 27, 47], LLMs exhibit proficiency in interpreting semantic data (e.g., activity trajectory data). This advantage significantly broadens the scope for incorporating a diverse array of data sources into generation processes, thereby enhancing the models' ability to understand and interact with complex, real-world scenarios in a more nuanced and effective manner.
- **Model Versatility.** Although other data-driven methods manage to learn such dynamic activity patterns for generation, their capacity is limited for generation under unseen scenarios. On the contrary, LLMs have shown remarkable versatility in dealing with unseen tasks, especially the ability to reason and decide based on available information [26]. This competence enables LLMs to offer a diverse and rational array of choices, making it a promising and flexible approach for modeling personal mobility patterns.

Despite these benefits, ensuring that LLMs align effectively with real-world situations continues to be a significant challenge [36]. This alignment is particularly crucial in the context of urban mobility, where the precision and dependability of LLM outputs are essential for the efficacy of any urban management derived from them. In this study, our aim is to address this challenge by investigating the following research questions: **RQ 1:** How can LLMs be effectively aligned with semantically rich data about daily individual activities? **RQ 2:** What are the effective strategies for achieving reliable and meaningful activity generation using LLM agents? **RQ 3:** What are the potential applications of LLM agents in enhancing urban mobility analysis?

To this end, our study employs LLM agents to infer activity patterns and motivation for personal activity generation tasks. While previous researches advocate habitual activity patterns and motivations as two critical elements for activity generation [17, 43], our proposed framework introduces a more interpretable and effective solution. By leveraging the capabilities of LLMs to process semantically rich datasets (e.g., personal check-in data), we enable a nuanced and interpretable simulation of personal mobility. Our methodology revolves around two phases: (1) activity pattern identification and (2) motivation-driven activity generation. In Phase 1, we leverage the semantic awareness of LLM agents to extract and identify self-consistent, personalized habitual activity patterns from historical data. In Phase 2, we develop two interpretable retrieval-augmented strategies that utilize the patterns identified in Phase 1. These strategies guide LLM agents to infer underlying daily motivations, such as evolving interests or situational needs. Finally, we instruct LLM agents to act as urban residents according to the obtained patterns and motivations. In this way, we generate their daily activities in a specific reasoning logic.

We evaluate the proposed framework using GPT-3.5 APIs over a personal activity trajectory dataset of Tokyo. The results demonstrate the capability of our framework to align LLM agents with semantically rich data for generating individual daily activities. The comparison with baselines,

such as attention-based methods [8, 22], adversarial learning methods [6, 42], and a diffusion model [46], underscores the advanced generative performance of our framework. The observation also suggests that our framework excels in reproducing temporal and spatio-temporal aspects of personal mobility generation and interpretable activity routines. Moreover, the application of the framework in simulating urban mobility under specific contexts, such as a pandemic scenario, reveals its potential to adapt to external factors and generate realistic activity patterns.

To the best of our knowledge, this study is *one of the pioneering works in developing an LLM agent framework for generating activity trajectory based on real-world data.* We summarize our contributions as follows: (1) We introduce a novel LLM agent framework for personal mobility generation featuring semantic richness. (2) Our framework introduces a self-consistency evaluation to ensure that the output of LLM agents aligns closely with real-world data on daily activities. (3) To generate daily activity trajectories, our framework integrates activity patterns with summarized motivations, with two interpretable retrieval-augmented strategies aimed at producing reliable activity trajectories. (4) By using real-world personal activity data, we validate the effectiveness of our framework and explore its utility in urban mobility analysis.

## 2   Related Work

### 2.1   Personal Mobility Generation

Activity trajectory generation offers a valuable perspective for understanding personal mobility. Based on vast call detailed records, Jiang et al. built a mechanistic modeling framework to generate individual activities in high spatial-temporal resolutions. Pappalardo and Simini employed Markov modeling to estimate the probability of individuals visiting specific locations. Besides, deep learning has become a robust tool for modeling the complex dynamics of traffic [15, 13, 44, 9, 21]. The primary challenge involves overcoming data-related obstacles such as randomness, sparsity, and irregular patterns [8, 43, 42, 20]. For example, Feng et al. proposed attentional recurrent networks to handle personal preference and transition regularities. Yuan et al. leveraged deep learning combined with neural differential equations to address the challenges of randomness and sparsity inherent in irregularly sampled activities for activity trajectory generation. Recently, Zhu et al. proposed to utilize a diffusion model to generate GPS trajectories.

### 2.2   LLM Agents in Social Science

Exploring how to treat LLMs as autonomous agents in specific scenarios leads to diverse and promising applications in social science [32, 36, 35, 10]. For instance, Park et al. established an LLM agent framework to simulate human behavior in an interactive scenario, demonstrating the potential of LLMs to model complex social interactions and decision-making processes. Moreover, the application of LLM agents in economic research has been explored, providing new insights into financial markets and economies [11, 19]. Extending beyond the realm of social sciences, Mao et al. adeptly utilized LLMs to generate driving trajectories in motion planning tasks. In the field of natural sciences, Williams et al. integrated LLMs with epidemic models to simulate the spread of diseases. These varied applications highlight the versatility and potential of LLMs to understand and model various real-world dynamics.

## 3   Methodology

We consider the generation of individual daily activity trajectories, each representing an individual's activities for the whole day. In addition, we focus on the urban context, where the activity trajectory of each individual is represented as a time-ordered sequence of location choices (e.g., POIs) [21]. This sequence is represented by $\{(l_0, t_0), (l_1, t_1), \ldots, (l_n, t_n)\}$, where each $(l_i, t_i)$ denotes the individual's location $l_i$ at time $t_i$.

By modeling individuals within an urban environment as LLM agents, we present LLMob, an LLM Agent Framework for Personal Mobility Generation, as illustrated in Figure 2. LLMob is based on the assumption that an individual's activities are primarily influenced by two principal factors: habitual activity patterns and current motivations. Habitual activity patterns, representing typical movement behaviors and preferences that indicate regular travel and location choices, are recognized

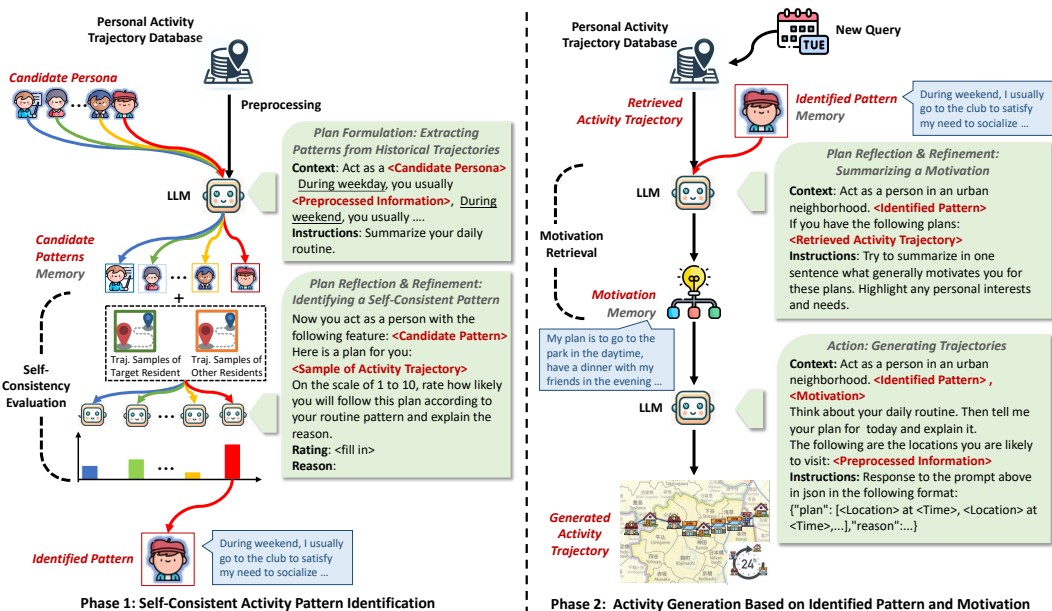

Figure 2: LLMob, the proposed LLM agent framework for personal Mobility generation.

as crucial information for inferring daily activities [31, 7, 30]. On the other hand, motivations relate to dynamic and situational elements that sway an individual's choices at any particular moment, such as immediate needs or external circumstances during a specific period. This consideration is vital for capturing and forecasting short-term shifts in mobility patterns [1, 43]. Moreover, by formulating prompts that assume specific events of concern, this framework allows us to observe the LLM agent's responses in a variety of situations.

To construct a pipeline for activity trajectory generation, we design an LLM agent with action, memory, and planning [33, 32]. Action specifies how an agent interacts with the environment and makes decisions. In LLMob, the environment contains the information collected from real-world data, and the agent acts by generating trajectories. Memory includes past actions that need to be prompted to the LLM to invoke the next action. In LLMob, memory refers to the patterns and motivations output by the agent. Planning formulates or refines a plan over past actions to handle complex tasks, with additional information optionally incorporated as feedback. In LLMob, we use planning to identify patterns and motivations, thereby handling the complex task of trajectory generation. Plan formulation, selection, reflection, and refinement [14] are employed in succession, and the agent keeps updating the action plan based on its observation [41]: The agent first formulates a set of activity plans by extracting candidate patterns from historical trajectories in the database. The agent then performs self-reflection through a self-consistency evaluation to pick the best pattern from the candidate patterns. With historical trajectories further retrieved from the database, the agent refines the identified pattern to a summarized motivation of daily activity, which is then jointly used with the identified pattern for trajectory generation. In addition to the above agentic components, we also suggest the personas of the agent, which can facilitate the LLM to simulate the diversity of real-world individuals [29].

## 3.1 Activity Pattern Identification

Phase 1 of LLMob focuses on identifying activity patterns from historical data. To effectively leverage the extracted activity patterns as essential prior knowledge for the generation of daily activities, we introduce the following two steps.

### 3.1.1 Pattern Extraction from Semantics and Historical Data

This step derives activity patterns based on activity trajectory data (e.g., individual check-in data). As illustrated in the left panel of Figure 2, this scheme consists of the following aspects: For each person, we start by specifying a candidate personas to the LLM agent, providing the inspiring foundation for subsequent activity pattern generation. This approach also encourages the diversity of the generated

activity patterns, as each candidate persona acts as a unique prior for the generation process (e.g., the significance of user clustering from activity trajectory data in producing meaningful distinctions has been demonstrated [24]). Meanwhile, we perform data preprocessing to extract key information from the extensive historical data. This involves identifying usual commuting distances, pinpointing typical start and end times and locations of daily trips, and concluding the most frequently visited locations of the person. It is important to note that these pieces of information are widely recognized as critical features in mobility analysis [17]. After the preprocessing procedure, both semantic elements with historical data are combined in the prompts, requiring the LLM agent to summarize the activity patterns for this person. By doing this, we set up a streamline to effectively bridge the gap between semantic persona characteristics and concrete historical activity trajectory data, which allows for a more personalized and interpretable representation of individual activities in one day. Moreover, we propose adding candidate personas to the prompt during candidate pattern generation to promote the diversity of the results. Without loss of generality, for each person, a set of $C$ ($C = 10$) candidate patterns, denoted as $\mathcal{CP}$, are generated according to the historical data and $C$ candidate personas, respectively. We provide the details of these candidate personas in Appendix C.4.

### 3.1.2 Pattern Evaluation with Self-Consistency

This step involves assessing the consistency of the candidate patterns to identify the most plausible one. We implement a scoring mechanism to evaluate the alignment of candidate patterns with historical data. To achieve this objective, we define a scoring function to gauge each candidate pattern $cp$ in the set $\mathcal{CP}$. This function evaluates $cp$ against two distinct sets of activity trajectories: the specific activity trajectories $\mathcal{T}_i$ of a targeted resident $i$ and the sampled activity trajectories from other residents $\mathcal{T}_{\sim i}$:

$$score_{cp} = \sum_{t \in \mathcal{T}_i} r_t - \sum_{t' \in \mathcal{T}_{\sim i}} r_{t'}, \tag{1}$$

where we design an evaluation prompt to ask the LLM to generate rating scores $r_t$ and $r_{t'}$. Specifically, the LLM agent is prompted to assess the degree of preference for a given trajectory based on the candidate pattern. Ideally, the LLM agent should assign a higher $r_t$ for data from the targeted resident and a lower $r_{t'}$ for data from other residents. This scheme essentially identifies the self-consistent pattern: the activity pattern derived from the activity trajectory data of the target user should be consistent with the data from this person during the evaluation. We provide the pseudo-code of the algorithm for Phase 1 of LLMob in Appendix A.

## 3.2 Motivation-Driven Activity Generation

In Phase 2 of LLMob, we focus on the retrieval of motivation and the integration of motivation and activity patterns for individual activity trajectory generation. Since the context length is limited for the LLMs, we can not expect that the LLMs can consume all the available historical information and give plausible output. Retrieval-augmented generation has been identified as a crucial factor in boosting the performance of LLM [37]. This enhancement provides additional information that aids LLM in more effectively responding to queries. While previous studies on activity generation mainly overlook the critical factors of macro temporal information (e.g., date) or specific scenarios (e.g., harsh weather) [42], we propose a more sophisticated activity generation which accounts for various conditions by taking advantage of the human-like intelligence of LLM. For instance, the activity trajectory at date $d$ can be inferred given the motivation of this date and the habitual activity pattern as:

$$\mathcal{T}_d = LLM(\mathcal{Motivation}, \mathcal{Pattern}). \tag{2}$$

This generation scheme instructs the LLM agent to simulate a designated individual according to a given activity pattern, and then meticulously generate an activity trajectory in accordance with the daily motivation. To obtain insightful and reliable motivations toward different aspects of data availability and sufficiency, two retrieval schemes are proposed. Notably, we considered them as two promising directions for designing solutions to real-world applications, rather than claming which is superior. The detail of each retrieval scheme is introduced as follows:

### 3.2.1 Evolving-based Motivation Retrieve

This scheme is related to the intuitive principle that an individual's motivation on any given day is influenced by her interests and priorities in preceding days [28]. Guided by this understanding, our approach harnesses the intelligence of the LLM agent to understand the behavior of daily activities and the underlying motivations. As illustrated in Figure 3, for a specific date $d$ for which we aim to generate the activity trajectory, we consider the activities of the past $k$ days ($k = \min(7, l)$, where $l$ is the maximum value such that the trajectory for date $d - l$ can be found in the database), and prompt the LLM

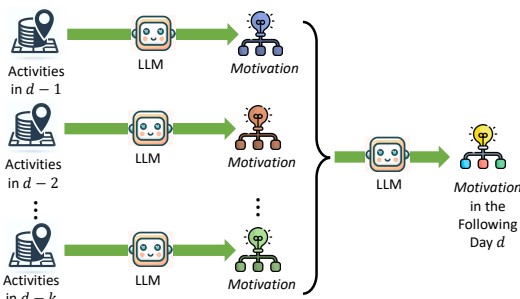

Figure 3: Evolving-based motivation retrieval.

agent to act as an urban resident based on the pattern identified in Section 3.1 and summarize $k$ motivations behind these activities. Using these summarized motivations, the LLM agent is further prompted to infer potential motivation for the target date $d$.

### 3.2.2 Learning-based Motivation Retrieval

In this scheme, we hypothesize that individuals tend to establish routines in their daily activities, guided by consistent motivations even if the specific locations may vary. For example, if someone frequently visits a burger shop on weekday mornings, this behavior might suggest a motivation for a quick breakfast. Based on this, it is plausible to predict that the same individual might choose a different fast food restaurant in the future, motivated by a similar desire for convenience and speed during their morning meal. We introduce a learning-based scheme to retrieve motivation from historical data. For each new date on which to plan activities, the only information available is the date itself. To use this clue for planning, we first formulate a relative temporal feature $z_{d_c, d_p}$ between a past date $d_p$ and the current date $d_c$. This feature captures various aspects, such as the gap between these two dates and whether they belong to the same month. Utilizing this setting, we train a score approximator $f_\theta(z_{d_c, d_p})$ to evaluate the similarity between any two dates. Notably, due to the lack of supervised signals, we employ unsupervised learning to train $f_\theta(\cdot)$. Particularly, a learning scheme based on contrastive learning [5] is established. For each trajectory of a resident, we can scan her other trajectories and identify similar (positive) and dissimilar (negative) dates according to a predefined similarity score. This similarity score is calculated between two activity trajectories $\mathcal{T}_{d_a}$ and $\mathcal{T}_{d_b}$ as:

$$sim_{d_a, d_b} = \sum_{t=1}^{N_d} \mathbf{1}_{(\mathcal{T}_{d_a}(t) = \mathcal{T}_{d_b}(t))} \text{ if } |\mathcal{T}_{d_a}| > t \text{ and } |\mathcal{T}_{d_b}| > t, \tag{3}$$

where $N_d$ is the total number of time intervals (e.g., 10 min) in one day. $\mathcal{T}_{d_a}(t)$ indicates the $t$th visiting location recorded in trajectory $\mathcal{T}_{d_a}$. Intuitively, there should be more shared locations in the similar trajectory pair. Thereafter, the positive pair is characterized by the highest similarity score, indicative of a greater degree of resemblance between the trajectories. Conversely, the negative pairs are marked by low similarity scores, reflecting a lesser degree of commonality. After obtaining the training dataset from these positive and negative pairs, we train a model to approximate the similarity score between any two dates by contrastive learning. This procedure involves the following steps:

1. For each date $d$, generate one positive pair $(d, d^+)$ and $k$ negative pairs $(d, d_1^-)$, ..., $(d, d_k^-)$ based on the similarity score and compute $z_{d,d^+}$, $z_{d,d_1^-}$, ..., $z_{d,d_k^-}$.

2. Forward the positive and negative pairs to $f_\theta(\cdot)$ to form:

$$\text{logits} = \left[ f_\theta(z_{d,d^+}), f_\theta(z_{d,d_1^-}), ..., f_\theta(z_{d,d_k^-}) \right]. \tag{4}$$

3. Adopt InfoNCE [25] as the contrastive loss function:

$$\mathcal{L}(\theta) = \sum_{n=1}^{N} - \log \left( \frac{e^{\text{logits}_i}}{\sum_{j=1}^{k+1} e^{\text{logits}_j}} \right)_n, \tag{5}$$

where $N$ is the batch size of the samples and $i$ indicates the index of the positive pair.

Upon training a similarity score approximation mode, it can be applied to access the similarity between any given query date and historical dates. This enables us to retrieve the most similar historical data, which is prompted to the LLM agent to generate a summary of the motivations prevalent at that time. By doing so, we can extrapolate a motivation relevant to the query date, providing a basis for the LLM agent to generate a new activity trajectory.

# 4 Experiments

## 4.1 Experimental Setup

**Dataset.** We investigate and validate LLMob over a personal activity trajectory dataset from Tokyo. This dataset was obtained through Twitter and Foursquare APIs and covers the data from January 2019 to December 2022. The time frame of this dataset is insightful as it captures typical daily life prior to the COVID-19 pandemic (i.e., normal period) and subsequent alterations during the pandemic (i.e., abnormal period). To facilitate a cost-efficient and detailed analysis for different periods, we randomly choose 100 users to model their individual activity trajectory at a 10-minute interval according to the number of available trajectories. Samples are shown in the following Table 1.

Table 1: Samples of personal activity data.

| UserID | Latitude | Longitude | Location Name | Category | Time |
|--------|----------|-----------|---------------|----------|------|
| 44673 | 35.008 | 139.015 | Convenience Store | Shop & Service | 2019-12-17 8:00 |
| 44673 | 35.009 | 139.018 | Ramen Restaurant | Food & Service | 2019-12-17 8:30 |
| 44673 | 35.004 | 139.060 | Italian Restaurant | Food & Service | 2019-12-17 11:20 |
| 44673 | 35.009 | 139.085 | Farmers Market | Shop & Service | 2019-12-17 14:20 |
| 44673 | 35.005 | 139.086 | Soba Restaurant | Food & Service | 2019-12-17 18:00 |

We utilize the category classification in Foursquare to determine the activity category for each location. We use 10 candidate personas (Appendix C.4) as a prior for subsequent pattern generation, which captures a diverse range of activity patterns within the data of this study. For the application to other datasets, this style of candidate patterns can be easily initialized using an LLM.

**Metrics.** The following characteristics related to personal activity are used to examine the generation: (1) **Step distance (SD)** [42]: The travel distance between each consecutive decision step within a trajectory is collected. This metric evaluates the spatial pattern of an individual's activities by measuring the distance between two consecutive locations in a trajectory. (2) **Step interval (SI)** [42]: The time gap between each consecutive decision step within a trajectory is recorded. This metric evaluates the temporal pattern of an individual's activities by measuring the time interval between two successive locations on an individual's trajectory. (3) **Daily activity routine distribution (DARD)**: For each decision step, a tuple $(t, c)$ is created, where $t$ represents the occurring time interval (e.g., from 0 to 144 in a day) and $c$ identifies the activity category based on the location visited at that step. A histogram is then constructed to represent the distribution of the collected tuples. This feature presents the patterns of individual activities characterized by activity type and timing (e.g., activity type, time). It provides insight into how activities are distributed over space and time and reflects semantic information such as habitual behavior. (4) **Spatial-temporal visits distribution (STVD)**: For each decision step, a tuple $(t, \text{latitude}, \text{longitude})$ is created, where $t$ represents the occurring time interval (e.g., from 0 to 144 in a day) and $\text{latitude}, \text{longitude}$ are the geographic coordinates of the location visited at that step. A histogram is subsequently built to represent the distribution of the collected tuples. This feature provides a granular perspective on the generated activities by assessing the spatial-temporal distribution of visited locations within each trajectory, including geographical coordinates and timestamps. It enables a detailed analysis of where and when activities occur.

After extracting the above characteristics from both the generated and real-world trajectory data, Jensen-Shannon divergence (JSD) is employed to quantify the discrepancy between them. Lower JSD is preferred.

**Methods.** LLMob is evaluated against: Markov-based mechanic model (MM) [27], an LSTM-based prediction model (LSTM) [12], two attention-based prediction models, including DeepMove [8] and STAN [22]. Within the domain of deep generative models, we select two adversarial learning frameworks, including TrajGAIL [6] and ActSTD [42], as well as a diffusion model, DiffTraj [46].

Table 2: Performance (JSD) of trajectory generation based on historical data. Lower is better. Winners and runners-up are marked in boldface and underline, respectively.

| Models | Normal Trajectory, Normal Data (# Generated Trajectories: 1497) | | | | Abnormal Trajectory, Abnormal Data (# Generated Trajectories: 904) | | | | Abnormal Trajectory, Normal Data (# Generated Trajectories: 3555) | | | |
|---|---|---|---|---|---|---|---|---|---|---|---|---|
| | SD | SI | DARD | STVD | SD | SI | DARD | STVD | SD | SI | DARD | STVD |
| MM [27] | 0.018 | 0.276 | 0.644 | 0.681 | 0.041 | 0.300 | 0.629 | 0.682 | 0.039 | 0.307 | 0.644 | 0.681 |
| LSTM [12] | 0.017 | 0.271 | 0.585 | 0.652 | 0.016 | 0.286 | 0.563 | 0.655 | 0.035 | 0.282 | 0.585 | 0.653 |
| DeepMove [8] | **0.008** | 0.153 | 0.534 | 0.623 | 0.011 | 0.173 | 0.548 | 0.668 | **0.013** | 0.173 | 0.534 | 0.623 |
| STAN [22] | 0.152 | 0.400 | 0.692 | 0.692 | 0.115 | 0.092 | 0.693 | 0.691 | 0.142 | 0.094 | 0.692 | 0.690 |
| TrajGAIL [6] | 0.128 | 0.058 | 0.598 | **0.489** | 0.133 | 0.060 | 0.604 | **0.523** | 0.332 | 0.058 | 0.434 | **0.428** |
| ActSTD [42] | 0.034 | 0.436 | 0.693 | 0.692 | 0.071 | 0.469 | 0.692 | 0.692 | 0.022 | 0.093 | 0.468 | 0.692 |
| DiffTraj [46] | 0.052 | 0.251 | 0.318 | 0.692 | **0.008** | 0.240 | 0.339 | 0.692 | 0.101 | 0.142 | 0.218 | 0.693 |
| LLMob-E | 0.053 | **0.046** | **0.125** | 0.559 | 0.056 | **0.043** | 0.127 | 0.615 | 0.062 | 0.056 | **0.117** | 0.536 |
| LLMob-E w/o $\mathcal{P}$ | 0.055 | 0.069 | 0.223 | 0.530 | 0.059 | 0.081 | 0.252 | 0.673 | 0.065 | 0.079 | 0.209 | 0.561 |
| LLMob-E w/o $\mathcal{SC}$ | 0.058 | 0.076 | 0.295 | 0.589 | 0.068 | 0.086 | 0.225 | 0.649 | 0.072 | 0.096 | 0.301 | 0.589 |
| LLMob-L | 0.049 | 0.054 | 0.136 | 0.570 | 0.057 | 0.051 | **0.124** | 0.609 | 0.064 | **0.051** | 0.124 | 0.531 |
| LLMob-L w/o $\mathcal{P}$ | 0.061 | 0.080 | 0.270 | 0.600 | 0.072 | 0.081 | 0.286 | 0.641 | 0.073 | 0.091 | 0.248 | 0.580 |
| LLMob-L w/o $\mathcal{SC}$ | 0.057 | 0.074 | 0.236 | 0.602 | 0.071 | 0.084 | 0.236 | 0.642 | 0.073 | 0.094 | 0.286 | 0.622 |
| LLMob w/o $\mathcal{M}$ | 0.059 | 0.078 | 0.264 | 0.590 | 0.066 | 0.080 | 0.274 | 0.633 | 0.074 | 0.090 | 0.255 | 0.563 |
| LLMob w/o $\mathcal{P}$ & $\mathcal{M}$ | 0.061 | 0.081 | 0.268 | 0.606 | 0.068 | 0.086 | 0.287 | 0.635 | 0.074 | 0.095 | 0.254 | 0.573 |

We use the source codes of the baselines provided by their respective authors and adapt them to our setting.

To achieve a balance between capability and cost efficiency, we employ GPT-3.5-turbo-0613 as the LLM core. We use "LLMob-E" to represent the proposal with the **evolving-based motivation retrieval** scheme, and "LLMob-L" to denote the framework that incorporates the **learning-based motivation retrieval** scheme (parameter settings in Appendix C.3). To validate the necessity of each module proposed, we conduct ablation studies with the following configurations: "LLMob w/o $\mathcal{P}$" denotes the framework generating trajectories without using the pattern (i.e., directly summarizing motivations from past trajectories). "LLMob w/o $\mathcal{M}$" denotes the framework without the motivation (i.e., directly generating trajectories with the identified pattern). "LLMob w/o $\mathcal{SC}$" denotes the framework without the self-consistency evaluation (in this case, a candidate pattern is randomly picked as the identified pattern). Furthermore, "LLMob w/o $\mathcal{P}$ & $\mathcal{M}$" represents the framework excluding both patterns and motivations.

## 4.2 Main Results and Analysis

**Generative Performance Validation (RQ 1, RQ 2).** The performance evaluation involves analyzing generation results in three distinct settings: (1) Generating normal trajectories based on normal historical trajectories in 2019, a period unaffected by the pandemic. (2) Generating abnormal trajectories based on abnormal historical trajectories in 2020, a year marked by the pandemic. (3) Generating abnormal trajectories in 2021 (pandemic) based on normal historical trajectories in 2019.

The results of these evaluations are detailed in the metrics reported in Table 2. Through the comparison, it can be observed that although LLMob may not excel in replicating spatial features (SD) precisely, it demonstrates superior performance in handling temporal aspects (DI). When considering spatial-temporal features (DARD and STVD), LLMob's performance is also competitive. In particular, LLMob achieves the best performance on DI and DARD for all three settings and is the runner-up on STVD. Baselines like DeepMove and TrajGAIL perform the best on SD and STVD, respectively, but become much less competitive when evaluated in other aspects. We suggest that the pronounced advantage of LLMob in terms of DARD (roughly 1/2 to 1/3 JSD compared to the best of baselines) can be attributed to the LLM agent's tendency to accurately replicate the motivation behind individual activity behaviors. For instance, an agent may recognize patterns like a person's habits to have breakfast in the morning, without being restricted to a specific restaurant. This phenomenon highlights the enhanced semantic understanding capabilities of the LLM agent.

**Exploring Utility in Real-World Applications (RQ 3).** We are interested in how LLMob can elevate the social benefits, particularly in the context of urban mobility. To this end, we propose an example of leveraging the flexibility and intelligence of LLM agents in understanding semantic information and simulating an unseen scenario. In particular, we enhance the original setup by incorporating an additional prompt to provide a context for the LLM agent, enabling it to plan activities during specific circumstances. For example, a "pandemic" prompt is as follows: *Now it is the pandemic period. The government has asked residents to postpone travel and events and to telecommute as much as possible.*

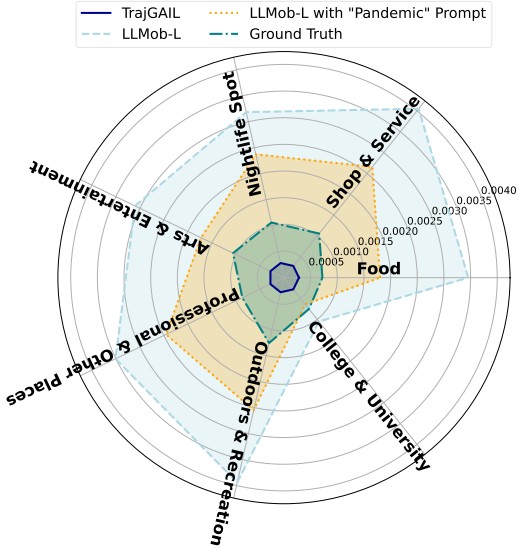

Figure 4: Daily activity frequency.

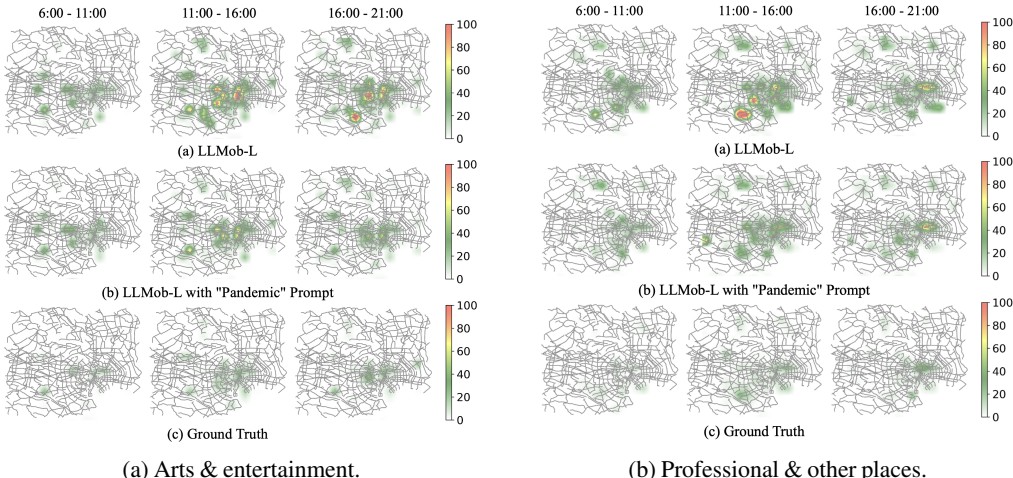

(a) Arts & entertainment.      (b) Professional & other places.

Figure 5: Activity heatmaps for the pandemic scenario.

By integrating the above prompt, we can observe the impact of external elements, such as the pandemic and the government's measures, on urban mobility and related social dynamics. We use the activity trajectory data during the pandemic (2021) as ground truth and plot the daily activity frequency in 7 categories in Figure 4. TrajGAIL, despite delivering the best STVD in Table 2, displays very low frequencies for all the categories, and fails to reflect the tendency of each category. In contrast, a comparison between LLMob-L and the one augmented with the pandemic prompt demonstrates the impact of external factors: there is a significant decrease in activity frequency with the pandemic prompt, which semantically discourages activities likely to spread the disease (e.g., food).

Additionally, from a spatial-temporal perspective, two major activities (e.g., *Arts & entertainment* and *Professional & other places*) are selected to observe the behavior, as shown in Figures 5a and 5b. These activities are particularly insightful as they encapsulate the impact of the pandemic on the work-life balance and daily routines of residents. Specifically, with the pandemic prompt, LLMob reproduces a more realistic spatial-temporal activity pattern. This enhanced realism in the generation is attributed to the integration of prior knowledge about the pandemic's effects and governmental responses, allowing the LLM agent to behave in a manner that aligns with actual behavioral adaptations. For instance, the reduction in *Arts & entertainment* activities reflects the closure of venues and social distancing guidelines, while changes in *Professional & other places*

activities indicate shifts toward remote work and the transformation of professional environments. Intuitively, prompting the LLM agent to generate activities based on various priors shows great potential in real-world applications. The utility of such a conditioned generative approach, coupled with the reliable generated results, can significantly alleviate the workload of urban managers. We suggest that this kind of workflow can simplify the analysis of urban dynamics and aid in assessing the potential impact of urban policies.

### 4.3 Ablation Studies

**Impact of Patterns.** In Table 2, by comparing LLMob with and without using patterns ("w/o $\mathcal{P}$"), we observe that the identified patterns consistently enhance the trajectory generation performance. The improvement on DARD is the most significant (reducing JSD by around 50%), showcasing the use of patterns is a key factor in capturing the semantics of daily activity. We provide example patterns in Appendix D.1 to show how the habitual behaviors of individuals are recognized by patterns.

**Impact of Self-Consistency Evaluation.** By comparing LLMob with and without self-consistency evaluation ("w/o $\mathcal{SC}$") in Table 2, we find that self-consistency is useful in all aspects, and its impact is the most significant on DARD, especially when generating abnormal trajectories from normal data, showcasing its effectiveness in processing semantics. We also observe that "w/o $\mathcal{SC}$" performs even worse than "w/o $\mathcal{P}$" in many cases, because in "w/o $\mathcal{SC}$", a candidate pattern is randomly picked for summarizing motivations, potentially introducing inconsistency to an individual's daily activity.

**Impact of Motivations.** We compare LLMob with and without motivations ("w/o $\mathcal{M}$"). As can been seen in Table 2, the impact of motivations is similar to that of patterns. By comparing to LLMob with both patterns and motivations removed ("w/o $\mathcal{P}$ & $\mathcal{M}$"), we observe that these two factors collectively lead to better performance. To show the motivations and the generated trajectories, we provide examples in Appendix D.2, where consistency between them can be observed.

**Impact of Motivation Retrieval Strategy.** We compare LLMob equipped with the two motivation retrieval strategies ("-E" and "-L"). Table 2 shows that no retrieval strategy always outperforms the other, though evolving-based retrieval wins in more cases (7 vs 5). Moreover, evolving-based retrieval is generally less sensitive to the removal of patterns or self-consistency evaluation, suggesting that resorting to the LLM to process historical trajectories is more robust than using contrastive learning.

## 5 Conclusion

**Contributions.** This study is believed to be the first personal mobility simulation empowered by LLM agents on real-world data. Our innovative framework leverages activity patterns and motivations to direct LLM agents in emulating urban residents, facilitating the generation of interpretable and effective individual activity trajectories. Extensive experimental studies based on real-world data are conducted to validate the proposed framework and demonstrate the promising capabilities of LLM agents to improve urban mobility analysis.

**Social Impacts.** Leveraging artificial intelligence to enhance societal benefits is increasingly promising, especially with the advent of high-capacity models such as LLMs. This study explores one of the potential avenues for applications using LLMs as reliable agents to simulate specific scenarios to assess the effects of external factors, such as pandemics and government policies. The introduced framework offers a flexible approach to enhance the reliability of LLMs in simulating urban mobility.

**Limitations.** In this study, we focused on modeling the activities of individual agents without considering interactions between them. As future work, we aim to extend this to a multi-agent scenario to capture interactions (e.g., where individuals may follow the activities of friends or family members). Given the challenges in collecting high-quality personal mobility data—many datasets lack completeness in capturing daily activities—we limited our comprehensive experimental evaluation to a single dataset. Furthermore, due to cost-efficiency considerations, only GPT-3.5 was fully evaluated. An additional analysis in a different city is provided in Appendix D.3, and Appendix D.4 includes supplementary evaluations using other LLMs.

## Acknowledgements

This work is supported by JSPS KAKENHI JP22H03903, JP23H03406, JP23K17456, JP24K02996, and JST CREST JPMJCR22M2.

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

# Appendix

## A   Algorithm Pseudo-Codes

The pseudo-code of the algorithm for activity pattern identification is given in Algorithm 1.

---

**Algorithm 1:** Activity Pattern Identification

**Input**    :Number of personas $C$, activity trajectories $\mathcal{T}$.
**Output**  :Best activity pattern $best\_pattern$.

1  Initialize: Empty candidate pattern set $\mathcal{CP}$;
2  Formulate initial pattern summarization prompt $p_1$ using prior information extracted from $\mathcal{T}$;
3  **for** $i = 1$ to $C$ **do**
4       Formulate prompt $p_1$ with prior information and persona $i$;
5       $cp \leftarrow \text{LLM}(p_1)$;
6       $\mathcal{CP} \leftarrow \mathcal{CP} \cup \{cp\}$;
7  Initialize a score dictionary $\mathcal{S}$ to store pattern scores;
8  **foreach** candidate pattern $cp$ in $\mathcal{CP}$ **do**
9       Initialize $score_{cp} = 0$;
10      **foreach** activity trajectory $t$ in $\mathcal{T}$ **do**
11          Formulate evaluation prompt $p_2$ for $cp$ and $t$;
12          $score_{cp} \leftarrow score_{cp} + \text{LLM}(p_2)$;
13      $\mathcal{S}[cp] \leftarrow score_{cp}$;
14 $best\_pattern \leftarrow \text{argmax}_{cp \in \mathcal{CP}} \mathcal{S}[cp]$;
15 **return** $best\_pattern$

---

## B   Prompts

**Pattern generation**

*Context: Act as a <INPUT 0> in an urban neighborhood, <INPUT 1> <INPUT 2>. There are also locations you sometimes visit: <INPUT 3>.*
*Instructions: Reflecting on the context given, I'd like you to summarize your daily activity patterns. Your description should be coherent, utilizing conclusive language to form a well-structured paragraph.*
*Text to Continue: I am <INPUT 0> in an urban neighborhood, <INPUT 1>,...*

where <INPUT 0> and <INPUT 1> are replaced by the candidate personas, and <INPUT 2> and <INPUT 3> will be replaced by the activity habits extracted from historical data. Specifically, we formulate the <INPUT 2> in the following format:

**<INPUT 2> format**

*During <weekend or weekday>, you usually travel over <distance> kilometers a day, you usually begin your daily trip at <time of first daily activity> and end your daily trip at <time of last daily activity>. you usually visit <location of first daily activity> at the beginning of the day and go to <location of last daily activity> before returning home.*

**Pattern evaluation**

*Now you act as a person with the follwing feature: <INPUT 0> Here is a plan for you. <INPUT 1> On the scale of 1 to 10, where 1 is the least possibility and 10 is the highest possibility, rate how likely you will follow this plan according to your routine pattern and explain the reason.*
*Rating: <fill in>*
*Reason:*

where <INPUT 0> is the candidate pattern, and <INPUT 1> is the daily activities plan for evaluation.

**Evolving-based motivation reterieval**

*Context: Act as a person in an urban neighborhood and describe the motivation for your activities. <INPUT 0> In the last <INPUT 1> days you have the following activities: <INPUT 2>*

*Instructions: Describe in one sentence your future motivation today after these activities. Highlight any personal interests and needs.*

where <INPUT 0> is replaced by the selected pattern, <INPUT 1> is replaced by the number of days from the date to plan activities, and <INPUT 2> is the historical activities corresponding to the chosen date.

**Learning-based motivation reterieval**

*Context: Act as a person in an urban neighborhood. <INPUT 0> If you have the following plans: <INPUT 1>*

*Instructions: Try to summarize in one sentence what generally motivates you for these plans. Highlight any personal interests and needs.*

where <INPUT 0> is replaced by the selected pattern, and <INPUT 1> Is replaced by the retrieved historical activities.

**Daily activities generation**

*Context: Act as a person in an urban neighborhood. <INPUT 0> Following is the motivation you want to achieve: <INPUT 1>*

*Instructions: Think about your daily routine. Then tell me your plan for today and exlpain it. The following are the locations you are likely to visit: <INPUT 2> Response to the prompt above in the following format: {"plan": [<Location> at <Time>, <Location> at <Time>,...], "reason":...}*

*Example: {"plan": [Elementary School*
*#125 at 9:10, Town Hall*
*#489 at 12:50, Rest Area*
*#585 at 13:40, Seafood Restaurant*
*#105 at 14:20] "reason": "My plan today is to finish my teaching duty in the morning and find something delicious to taste."}*

where <INPUT 0> is replaced by the selected pattern, <INPUT 1> is replaced by the retrieved motivation, and <INPUT 2> is replaced by the most frequently visited locations.

**Pandemic**

*Now it is the pandemic period. The government has asked residents to postpone travel and events and to telecommute as much as possible.*

## C  Experimental Setup

### C.1  Data processing

All the data is obtained through the Twitter and Foursquare APIs and is already anonymized to remove any personally identifiable information before analysis. The detailed process is as follows:

1. **Filtering Incomplete Data**
   Users with missing check-ins for a specific year were filtered out.
2. **Excluding Non-Japan Check-ins**
   Check-ins that occurred outside of Japan were removed.
3. **Inferring Prefecture from GPS Coordinates**
   Prefectures were inferred based on the latitude and longitude data of check-ins.

4. **Assigning Prefecture**
   Users were assigned to a prefecture based on their primary check-in location; for example, users whose top check-in location is Tokyo were categorized as belonging to Tokyo.
5. **Removing Sudden-Move Check-ins**
   Check-ins showing abrupt, unrealistic location changes, such as from Tokyo to the United States within a short time frame, were deleted to remove data drift, following the criteria proposed by [40].
6. **Anonymizing Data**
   Real user IDs and geographic location names were anonymized. Only category information of geographic locations was kept, and latitude and longitude coordinates were converted into IDs before being input into the model.

## C.2 Environment

We leverage the GPT API to conduct our generation studies. Specifically, we use the gpt-3.5-turbo-0613 version of the API, which is a snapshot of GPT-3.5-turbo from June 13th, 2023. The experiments were carried out on a server with the following specifications:

- **CPU**: AMD EPYC 7702P 64-Core Processor
  - **Architecture**: x86_64
  - **Cores/Threads**: 64 cores, 128 threads
  - **Base Frequency**: 2000.098 MHz
- **Memory**: 503 GB
- **GPUs**: 4 x NVIDIA RTX A6000
  - **Memory**: 48GB each

## C.3 Learning-Based Motivation Retrieval

For the learning-based motivation retrieval, the score approximator is parameterized using a fully connected neural network with the following architecture:

Table 3: Architecture of the score approximator.

| Layer | Input Size | Output Size | Notes |
|---|---|---|---|
| Input Layer | 3 | 64 | Linear |
| Activation | - | - | ReLU |
| Output Layer | 64 | 1 | Linear |

We include the day of the year for the query date, whether it shares the same weekday as the reference date, and whether both the query and reference dates fall within the same month as input features. Settings for the learning process are as follows: Adam [18] is used as the optimizer, batch size is 64, learning rate is 0.002, and the number of negative samples is 2.

## C.4 Personas

We use 10 candidate personas as a prior infromation for subsequent pattern generation, as shown in Table 4.

# D  Additional Experimental Results

## D.1 Examples of Identified Patterns

The patterns are extracted and identified during the first phase in our framework. We report some examples of the identified patterns in our experiments as follows, which correspond to the 10 personas in Table 4.

Table 4: Suggested personas and corresponding descriptions.

**Student**: typically travel to and from educational institutions at similar times.

**Teacher**: typically travel to and from educational institutions at similar times.

**Office worker**: have a fixed morning and evening commute,
often heading to office districts or commercial centers.

**Visitor**: tend to travel throughout the day,
often visit attractions, dining areas, and shopping districts.

**Night shift worker**: might travel outside of standard business hours,
often later in the evening or at night.

**Remote worker**: may have non-standard travel patterns,
often visit coworking spaces or cafes at various times.

**Service industry worker**: tend to travel throughout the day, often visit attractions,
dining areas, and shopping districts.

**Public service official**: often work in shifts,
leading to varied travel times throughout the day and night.

**Fitness enthusiast**: often travel early in the morning, in the evening,
or on weekends to fitness centers or parks.

**Retail employee**: travel patterns might include shifts that
start late in the morning and end in the evening.

---

**Pattern of a student**

You are a student in this urban neighborhood. You typically travel to and from educational institutions at similar times. Your daily routine as a student in an urban neighborhood revolves around traveling to and from educational institutions. On weekdays, you cover a distance of over 10 kilometers, starting your daily trip at 12:00 and concluding it at 20:20. At the beginning of the day, you usually visit Park#2457 and then head to Grocery Store#648 before returning home. Weekends follow a similar pattern, with you traveling over 10 kilometers a day, but starting your daily trip at 13:40 and ending it at 19:20. Like on weekdays, you begin by visiting Park#2457 and then proceed to Grocery Store#648 before heading back home. Throughout the week, you have specific places You visit at fixed times, such as Grocery Store#648 at 20:00, Coffee Shop#571 at 09:30, Park#2457 at 08:30, Irish Pub#21 at 20:00, and Bookstore#313 at 16:00. Additionally, there are other locations You occasionally visit, including Chinese Restaurant#168, Supermarket#802, Park#4719, Tea Room#530, Bookstore#336, Pastry Shop#240, Park#2898, Discount Store#807, and Electronics Store#530. By visiting educational institutions at similar times, you ensure punctuality and consistency in your studies. Additionally, your visits to the park and various stores serve as a way to relax, replenish supplies, and indulge in leisure activities. Overall, your daily routine is structured to fulfill both your educational and personal requirements efficiently.

## Pattern of a teacher

You are a teacher in this urban neighborhood. You typically travel to and from educational institutions at similar times. Your daily routine as a teacher in an urban neighborhood revolves around your regular travel to and from educational institutions. On weekdays, you cover a distance of over 60 kilometers a day, starting your journey at 11:50 in the morning and concluding it at 17:50 in the evening. At the beginning of the day, you usually visit Rest Area#1533 before heading to the Housing Development#101, where you spend a significant amount of time before returning home. During weekends, your travel distance reduces to 50 kilometers a day, with your daily trip commencing at 11:20 and ending at 18:00. On weekends, you follow a different routine, starting your day by visiting Shopping Mall#1262 and then proceeding to Motorcycle Shop#149 before heading back home. Additionally, there are certain locations You occasionally visit, such as Convention Center#101, Food Court#559, Motorcycle Shop#354, and Sports Bar#56, among others. Your daily routine is motivated by the need to fulfill your teaching responsibilities and ensure that you are present at the educational institutions You serve. The specific locations You visit, whether it's the Rest Area#1533 or Shopping Mall#1262, play a role in providing you with necessary resources, relaxation, or opportunities to engage in personal interests. Overall, your routine is structured to maintain a balance between professional commitments and personal needs.r relevant places, you can effectively serve the needs of the urban neighborhood and contribute to its smooth functioning.

## Pattern of a office worker

You are a office worker in this urban neighborhood. You have a fixed morning and evening commute, often heading to office districts or commercial centers. Your daily routine as an office worker in an urban neighborhood is quite structured. On weekdays, you travel over 30 kilometers a day, starting your daily trip at 06:50 and ending it at 20:20. Your routine usually begins with a visit to Platform#212, followed by heading to Soba Restaurant#955 before returning home. You also have certain places You visit at specific times, such as Hospital#1255 at 10:30, Platform#212 at 06:30, Soba Restaurant#955 at 22:30, Public Bathroom#76 at 08:30, and Platform#1068 at 05:00. During weekends, your daily travel distance increases to over 40 kilometers, and you start your trip at 07:00, ending it at 20:10. However, the pattern remains the same, starting with a visit to Platform#212 and then going to Soba Restaurant#955 before returning home.

## Pattern of a visitor

You are a visitor in this urban neighborhood. You tend to travel throughout the day, often visit attractions, dining areas, and shopping districts. Your daily routine as a visitor in this urban neighborhood involves traveling extensively and exploring various attractions, dining areas, and shopping districts. On weekdays, you cover a distance of over 40 kilometers, starting your day at 11:00 and concluding it at 17:10. You have a consistent pattern of visiting Ramen Restaurant#4841 at the beginning of the day and then heading to Town#373 before returning home. Weekends are slightly different, with a shorter distance of around 30 kilometers covered. You begin your day at 14:00 and end it at 16:50. During weekends, you start by visiting Ramen Restaurant#3773 and then proceed to Town#373 before heading back. There are certain locations that you frequently visit, such as Arcade#929 at 13:30, Bowling Alley#306 at 11:00, Arcade#408 at 12:00, and Grocery Store#2094 at 12:30. Additionally, you sometimes visit other places like Electronics Store#562, Comic Shop#5, Video Game Store#8, and many others. Your routine is motivated by your desire to explore and experience the various attractions, cuisines, and shopping opportunities available in this urban neighborhood. You find joy in discovering new places, trying different foods, and immersing yourself in the vibrant atmosphere of this bustling area.

## Pattern of a night shift worker

You are a night shift worker in this urban neighborhood. You travel to work in the evening and return home in the early morning. Your daily routine as a night shift worker in an urban neighborhood revolves around traveling a considerable distance. On weekdays, you cover over 70 kilometers a day, starting your daily trip at 09:20 and concluding it at 22:30. The routine begins with a visit to Toll Booth#194, followed by a stop at Supermarket#3823 before heading home. During weekends, the pattern remains the same, with the only difference being that you commence your journey at 10:30 and finish at 22:20. On these days, you visit Toll Booth#812 first and then head to Cafe#962 before returning home. In addition to these regular stops, there are occasional visits to various locations such as Record Shop#81, Lake#600, and Factory#495, among others. The motivation behind this routine is to ensure that you are well-prepared for your night shift, starting the day by taking care of essential tasks like stopping at toll booths and getting groceries. These routines help you maintain a sense of order and efficiency in your daily life, ensuring that you are ready for work and able to relax and enjoy your free time.

## Pattern of a remote worker

You are a remote worker in this urban neighborhood. You may have non-standard travel patterns, often visit coworking spaces or cafes at various times. Your daily routine as a remote worker in an urban neighborhood involves traveling a considerable distance, both during weekdays and weekends. On weekdays, you typically travel over 40 kilometers a day, starting your daily trip at 14:00 and ending it at 19:20. Your routine usually begins with a visit to Convention Center#101, followed by a stop at Supermarket#1593 before returning home. During weekends, your travel distance is slightly less, around 20 kilometers a day, and you begin your daily trip at 11:50, concluding it at 16:30. On weekends, you usually start by visiting Discount Store#884 and then head to Supermarket#1689 before returning home. Additionally, you have specific places You visit at certain times, including Supermarket#1593 at 17:00, Hobby Shop#516 at 13:30, Shopping Mall#1262 at 15:00, Hobby Shop#168 at 14:00, and Exhibit#461 at 11:30. Occasionally, you also visit other locations such as Shopping Mall#1073, Bookstore#14, Shrine#2783, and many more. Your motivation for this routine is to have a flexible work environment, utilizing coworking spaces and cafes, while also fulfilling your daily needs and exploring different places within the urban neighborhood.

## Pattern of a service industry worker

You are a service industry worker in this urban neighborhood. You might travel outside of standard business hours, often later in the evening or at night. Your daily routine as a service industry worker in an urban neighborhood is quite busy and revolves around your work schedule. On weekdays, you travel over 10 kilometers a day, starting your daily trip at 07:20 and ending it at 20:40. The day usually begins with a visit to Historic Site#2176, followed by a stop at Convenience Store#3385 before returning home. During weekends, your travel distance decreases to 0 kilometers a day, starting your daily trip at 09:20 and ending it at 20:10. Similar to weekdays, you start your day by visiting Historic Site#2176 and then go to Public Art#99 before returning home. Additionally, there are certain locations You sometimes visit, including various convenience stores, restaurants, shopping malls, and other establishments. Your motivation for this routine is primarily driven by your work commitments and the need to fulfill your responsibilities as a service industry worker. It is essential for you to visit specific places, such as convenience stores and historic sites, to ensure You have the necessary supplies and maintain a well-rounded understanding of the neighborhood. Overall, this routine enables you to efficiently navigate your urban neighborhood and fulfill your professional obligations.

## Pattern of a public service official

You are a public service official in this urban neighborhood. You often work in shifts, leading to varied travel times throughout the day and night.Your daily routine as a public service official in an urban neighborhood revolves around your shifts, which result in different travel times. On weekdays, you typically cover over 60 kilometers a day, starting your daily trip at 12:50 and concluding it at 19:00. At the beginning of the day, you make it a point to visit Convenience Store#3042, and before heading home, you stop by Convenience Store#3702. During the weekends, you travel around 50 kilometers daily, commencing your journey at 10:00 and wrapping it up at 18:30. On weekends, your routine involves visiting Platform#511 in the morning and then heading to Platform#670 before returning home. Additionally, there are a few locations You occasionally visit, such as Platform#1135, Home Service#244, and Convenience Store#6014, among others. The motivation behind your daily routine is to ensure that you cover the necessary ground, making essential stops at various locations to fulfill your duties as a public service official. By visiting convenience stores, platforms, home services, and other relevant places, you can effectively serve the needs of the urban neighborhood and contribute to its smooth functioning.

## Pattern of a fitness enthusiast

You are a fitness enthusiast in this urban neighborhood. You often travel early in the morning, in the evening, or on weekends to fitness centers or parks. Every weekday, you travel over 70 kilometers a day, starting your daily trip at 10:30 and ending it at 19:50. Your routine begins with a visit to Toll Booth#34, where you kickstart your day. After that, you head to Recreation Center#18 to engage in your fitness activities before returning home. On weekends, your daily travel distance is slightly less, around 60 kilometers. You start your trips at 12:40 and end them at 21:00. The weekend routine starts with a visit to Convenience Store#5940, where you grab some essentials for the day. Then, you head to Convenience Store#8965 before finally returning home. Throughout the week, you also occasionally visit other locations such as Tunnel#1307, Event Space#104, Shopping Mall#217 and #399, and various toll booths. Your motivation for this daily routine is your passion for fitness and maintaining a healthy lifestyle. You prioritize visiting fitness centers or parks during your trips to ensure that you have dedicated time for exercise. Additionally, you make sure to visit convenience stores for any necessary supplies and toll booths for smooth travel. Overall, your routine revolves around staying active, exploring different locations, and ensuring a well-rounded fitness experience.

## Pattern of a retail employee

You are a retail employee in this urban neighborhood. Your travel patterns might include shifts that start late in the morning and end in the evening. Every weekday, you embark on a daily journey that spans over 50 kilometers. Your routine begins at 09:30, and you conclude your travels at 17:00. To kickstart your day, you always make a point to visit Soba Restaurant#2105, relishing in their delicious offerings. Before heading home, you make a stop at Indian Restaurant#885, savoring their delectable cuisine. On weekends, your daily travel distance decreases slightly to 40 kilometers. You commence your excursions at 09:10 and wrap them up at 17:20. Your first stop on weekends is Hot Spring#205, where you indulge in relaxation and rejuvenation. Before returning home, you make a detour to Department Store#399, exploring the vast array of products they offer. Additionally, there are several other locations You occasionally visit, such as pharmacies, history museums, shrines, shopping malls, and more. Your motivation for this routine stems from your desire for variety and exploration. By visiting different establishments and places, you are able to experience diverse cuisines, immerse yourself in local culture, and find moments of serenity amidst the bustling city life.

## D.2 Examples of Retrieved Motivations and Corresponding Generated Trajectories

The retrieved motivations inspire the agent to plan daily activities that closely align with its specific needs. Here are some examples of retrieved motivations and the corresponding daily activities generated in our experiments.

---

**Example 1**

**Motivation:** Your motivations are primarily driven by your desire for entertainment and relaxation, as you enjoy going to the bowling alley and donburi restaurant during late hours, visiting the supermarket and rest area to fulfill your daily needs and take breaks.

**Daily Activity Trajectory Generated:** Activities at 2019-12-27: Print Shop (35.697, 139.404) at 09:10:00, Entertainment Place (35.733, 139.434) at 10:10:00, Donburi Restaurant (35.730, 139.435) at 10:30:00, Platform (35.733, 139.435) at 10:50:00, Convenience Store (35.721, 139.397) at 11:20:00, Platform (35.729, 139.392) at 21:30:00, Donburi Restaurant (35.699, 139.412) at 22:10:00, Convenience Store (35.697, 139.416) at 22:40:00.

---

**Example 2**

**Motivation:** Your motivations revolve around fulfilling basic needs such as food, convenience, relaxation, and personal interests, including a fondness for Japanese cuisine, maintaining a healthy lifestyle, and enjoying social gatherings in the evening.

**Daily Activity Trajectory Generated:** Activities at 2019-12-25: Pet Store (35.699, 139.404) at 08:10:00, Convenience Store (35.697, 139.416) at 13:10:00, Gym (35.697, 139.412) at 13:20:00, Massage Studio (35.698, 139.413) at 14:50:00, Electronics Store (35.699, 139.414) at 15:50:00, Fast Food Restaurant (35.697, 139.413) at 17:10:00, Platform (35.698, 139.415) at 18:10:00, Convenience Store (35.701, 139.393) at 19:10:00.

---

**Example 3**

**Motivation:** Your motivations in the urban neighborhood are primarily driven by the need for sustenance and convenience, ensuring personal safety and security, fulfilling work responsibilities, and satisfying personal preferences for food.

**Daily Activity Trajectory Generated:** Activities at 2019-12-30: Park (35.779, 139.491) at 06:40:00, Convenience Store (35.831, 139.519) at 07:10:00, Discount Store (36.356, 139.064) at 10:40:00, Assisted Living (36.338, 139.048) at 12:00:00, Bank (36.361, 139.059) at 12:10:00, Home Service (36.352, 139.050) at 15:30:00, Sushi Restaurant (36.373, 139.061) at 19:40:00, Hot Spring (36.340, 139.052) at 20:30:00.

---

**Example 4**

**Motivation:** Your motivations revolve around fulfilling basic needs such as food, convenience, relaxation, and personal interests, including a fondness for Japanese cuisine, maintaining a healthy lifestyle, and enjoying social gatherings in the evening.

**Daily Activity Trajectory Generated:** Activities at 2019-12-29: Park (35.779, 139.491) at 08:20:00, Event Space (35.630, 139.790) at 10:40:00, Convention Center (35.629, 139.795) at 12:20:00, Convenience Store (35.647, 139.817) at 13:00:00, Soba Restaurant (35.643, 139.820) at 13:10:00, Sporting Goods Shop (35.695, 139.762) at 14:50:00, Kushikatsu Restaurant (35.778, 139.495) at 16:40:00.

> **Example 5**
>
> **Motivation:** Your motivations revolve around your love for exploring different cuisines, seeking convenience and relaxation through internet cafes, and fulfilling your personal needs by visiting the grocery store.
>
> **Daily Activity Trajectory Generated:** Activities at 2019-12-31: Internet Cafe (35.723, 140.091) at 07:10:00, Donburi Restaurant (35.581, 140.132) at 12:40:00, Japanese Restaurant (35.369, 140.306) at 17:50:00, Rest Area (35.556, 140.208) at 18:30:00, Supermarket (35.865, 140.024) at 19:40:00.

### D.3 Experiment on Osaka Data

We conducted an experiment based on the data collected in Osaka, Japan. We generated 537 trajectories based on the 2102 daily activity trajectories from 30 persons. The results are reported as follows, where **LLMob-L/E** are ours and **DiffTraj** and **TrajGAIL** are the best-performing baseline methods.

Table 5: Comparison of models based on various metrics on Osaka data

| Model | SD | SI | DARD | STVD |
|---|---|---|---|---|
| LLMob-L | 0.035 | 0.021 | 0.141 | 0.391 |
| LLMob-E | 0.030 | 0.018 | 0.121 | 0.380 |
| DiffTraj | 0.080 | 0.177 | 0.406 | 0.691 |
| TrajGAIL | 0.281 | 0.063 | 0.525 | 0.483 |

### D.4 Experiment on different LLMs

We conducted experiments for setting (1) using different LLMs (**GPT-4o-mini** and **Llama 3-8B**). The results are reported as follows:

Table 6: Results of experiments using different LLMs

| Model | SD | SI | DARD | STVD |
|---|---|---|---|---|
| LLMob-L (GPT-3.5-turbo) | 0.049 | 0.054 | 0.136 | 0.570 |
| LLMob-L (GPT-4o-mini) | 0.049 | 0.055 | 0.141 | 0.577 |
| LLMob-L (Llama 3-8B) | 0.054 | 0.063 | 0.119 | 0.566 |
| LLMob-E (GPT-3.5-turbo) | 0.053 | 0.046 | 0.125 | 0.559 |
| LLMob-E (GPT-4o-mini) | 0.041 | 0.053 | 0.211 | 0.531 |
| LLMob-E (Llama 3-8B) | 0.054 | 0.059 | 0.122 | 0.561 |

We observe competitive performance of our framework when other LLMs are used. In particular, **GPT-4o-mini** is the best in terms of the spatial metric (SD); **GPT-3.5-turbo** is the best in terms of the temporal metric (SI). **Llama 3-8B** is overall the best when spatial and temporal factors are evaluated together (DARD and STVD). Such results demonstrate the robustness of our framework across different LLMs.

