# OpenReview forum: "Large Language Models as Urban Residents: An LLM Agent Framework for Personal Mobility Generation"
_NeurIPS.cc/2024/Conference — NeurIPS 2024 poster_

### Official Review · Reviewer_c5UM · 2024-06-24

**Soundness:** 3
**Presentation:** 4
**Contribution:** 3
**Rating:** 6
**Confidence:** 4

**Summary:**

The paper proposes an LLM-based personal activity generation by utilizing existing OpenAI LLM through API, named as LLMob. The problem that authors are time and important research topic in the area of human mobility simulation. While existing generative solely trained on the given dataset, LLM agent has already trained internet-wide dataset that allows it to have possibility of generating human-like trajectories. The LLMob generates activity trajectories in two stages: the first stage identifies (learns) activity patterns with respect to different personas introduced by the authors, and in the second stage a synthetic version of the activity sequences generated by LLM agent. The experiments are performed on an activity dataset, and performance is benchmarked by multiple generative models.

**Strengths:**

The paper is a good example of an agentic-LLM, showcasing the powerful capabilities of LLMs in human mobility modeling. This application is significant as it uses LLMs' data processing and predictive abilities to improve our understanding of human movement patterns, impacting urban planning, transportation, and public health. The paper is well-structured, with a clear and coherent flow, and features a promising presentation that enhances comprehension.

**Weaknesses:**

These are the set of  weaknesses of the paper from my perspective:
1-) The authors performed experiments on one dataset. It might be interesting to examine the performance with dataset from a different region.
2-) The model solely relies on the pre-trained LLM models. While the LLM capabilities and power are increasing, which is good for this work, separate fine-tuning might be required for concrete human mobility modeling.
3-) The motivation retrieval part is not clear, especially the learning-based motivation retrieval.

**Questions:**

While Phase 1 and Phase 2 make sense. I have concerns about how much information was carried from Phase 1 to Phase 2. The following are the questions that can help me to understand the details:

1-) Did you utilize the whole dataset in Phase 1?
2-) How are the trajectories and personas sampled in Phase 2?
3-) How do the generated activities match with the input trajectories? Does LLM memorize exact activity sequences?
4-) LLMs are known for hallucination. Within generated activities, the LLM agent generates an activity name and location. How accurate is the LLM agent for the generated activities? For instance, when an activity says Japanese Restaurant (35.369, 140.306), does the GPS location have a Japanese Restaurant? I expect the generated trajectories to require extensive post-processing.
5-) Have you tried your model on a different dataset? Experiments are limited to one dataset, which limits generalizability.

**Limitations:**

I found two limitations in this paper and would love to hear authors responses on these:

1-) While LLMs are generally powerful, they often struggle to generalize in domain-specific applications. For a robust human mobility simulation, LLM agents might require a fine-tuning strategy to achieve more generalizable performance.

2-) The generated locations in the study are limited to those specified in the prompt (If they do not hallucinate). Since the model also takes into account the locations a person is likely to visit, as illustrated in Figure 2, the degree of synthetic realism in the generated trajectories raises questions.

3-) This approach also does not fully address privacy concerns associated with mobility trajectories as mentioned in the introduction, the underlying patterns and personal data might still be exposed.

---

> ### Author Rebuttal · Authors · 2024-08-01
>
> Thank you for your insightful review and acknowledgment of our work. We would like to address the concerns as follows.
>
> > **Question 1** *Did you utilize the whole dataset in Phase 1?*
>
> **Answer**: Phase 1 is targeted at identifying the persona of each person. For the sake of fair comparison, for each targeted person, we used 80% of the available data of this person in this phase.
>
> > **Question 2** *How are the trajectories and personas sampled in Phase 2?*
>
> **Answer**: In Phase 1, we start with 10 predefined personas (Table 4) representing diverse urban lifestyles. For each individual, we prompt the LLM to refine these personas based on information extracted from their historical data, resulting in 10 candidate personas. The final persona is then determined through our self-consistency evaluation process, which assesses how well each candidate aligns with the individual's actual behavior patterns.
>
> In Phase 2, the identified persona from Phase 1 is incorporated into the prompt as an element. For trajectory generation, we employ two schemes:
>
> 1. Evolving-based scheme: We sample the individual's **latest one-week historical trajectories** to inform the LLM about recent activity patterns and motivations.
>
> 2. Learning-based scheme: We use a learning model (using 80% of available data) to identify **similar dates**, allowing the LLM to draw insights from historically similar contexts.
>
> This approach allows us to generate personalized, context-aware trajectories that strike a balance between consistent behavior patterns and daily variations in motivation and circumstances.
>
> > **Question 3** *How do the generated activities match with the input trajectories? Does LLM memorize exact activity sequences?*
>
> **Answer**: During generation, the prompt consists of the identified pattern of the targeted person and the retrieved motivation (Please see Appendix B: Page 14, Line 515). We are not providing specific trajectories to the LLM but the retrieved motivations, thus the LLM will **not** need to memorize exact activity sequences.
>
> > **Question 4** *LLMs are known for hallucination. Within generated activities, the LLM agent generates an activity name and location. How accurate is the LLM agent for the generated activities? For instance, when an activity says Japanese Restaurant (35.369, 140.306), does the GPS location have a Japanese Restaurant? I expect the generated trajectories to require extensive post-processing.*
>
> **Answer**: We evaluated the generated activities by checking whether the generated one has appeared in the Tokyo area. We found that all the locations generated from the model exist in the area.
>
> In addition, (35.369, 140.306) is the real location of a Japanese restaurant ("やきとり 福仙", an authentic Japanese yakitori restaurant).
>
> > **Question 5** *Have you tried your model on a different dataset? Experiments are limited to one dataset, which limits generalizability.*
>
> **Answer**: We conducted an experiment based on the data collected in Osaka, Japan. We generated 537 trajectories based on the 2102 daily activity trajectories from 30 persons. The results are reported as follows, where LLMob-L/E are ours and DiffTraj and TrajGAIL are the best-performing baseline methods.
>
> | Model          | SD        | SI        | DARD      | STVD      |
> |----------------|-----------|-----------|-----------|-----------|
> | LLMob-L        | 0.035     | 0.021     | 0.141     | 0.391     |
> | LLMob-E        | 0.030     | 0.018     | 0.121     | 0.380     |
> | DiffTraj       | 0.080     | 0.177     | 0.406     | 0.691     |
> | TrajGAIL       | 0.281     | 0.063     | 0.525     | 0.483     |
>
> The above results demonstrate that our framework can maintain superior performance in another city. In Section 5 - Limitations, we acknowledged that it is challenging to collect sufficient data from different areas, which limits our ability to conduct more extensive experiments. Additionally, we note that the model's generalization ability can be also demonstrated by its performance across different scenarios in our original experiments, such as under normal periods and under pandemic periods.

---

> ### Author Response · Authors · 2024-08-07
> **Rebuttal by Authors**
>
> > **Limitation 1** *While LLMs are generally powerful, they often struggle to generalize in domain-specific applications. For a robust human mobility simulation, LLM agents might require a fine-tuning strategy to achieve more generalizable performance.*
>
> **Answer**: We agree that fine-tuning the LLM for a specific domain may deliver better results. However, this needs careful design of instruction-tuning data, and the tuning may incur significant training costs, considering that we are using the GPT API. Indeed, we have tried fine-tuning GPT-3.5. After comparison, we chose the current approach for a better trade-off.
>
> > **Limitation 2** *The generated locations in the study are limited to those specified in the prompt (If they do not hallucinate). Since the model also takes into account the locations a person is likely to visit, as illustrated in Figure 2, the degree of synthetic realism in the generated trajectories raises questions.*
>
> **Answer**: We appreciate the reviewer's observation. Our approach to specifying likely locations is grounded in research by [1], which shows that individuals typically have characteristic geographical ranges for daily activities.
>
> [1] Alessandretti et al. The scales of human mobility. Nature, 2020, 587(7834): 402-407.
>
> However, our method does not limit generated locations to those specified. The LLM can reason beyond these locations, potentially introducing new, plausible ones based on the context and motivation. This balances realistic patterns with diversity.
>
> The use of location priors enhances generation efficiency and grounds outputs in realistic spatial contexts. Our evaluations, comparing generated trajectories to real-world data, demonstrate that our method produces patterns closely matching real-world data across various metrics.
>
> In future work, we plan to explore dynamically expanding the set of potential locations based on evolving trajectory contexts, further enhancing realism and flexibility.
>
> > **Limitation 3** *This approach also does not fully address privacy concerns associated with mobility trajectories as mentioned in the introduction, the underlying patterns and personal data might still be exposed.*
>
> **Answer:** We suppose that the reviewer's concern comes from the data exposed to the OpenAI.
>
> To address this concern, we conducted experiments using Llama 3-8B on a local GPU to ensure data security. The results are reported as follows (we also tested GPT-4o-mini):
>
> | Model          | SD | SI | DARD | STVD |
> |----------------|----------|-----------|--------|--|
> | LLMob-L (GPT-3.5-turbo)      |   0.049   |  0.054    |   0.126  |   0.570 |
> | LLMob-L (GPT-4o-mini)      |   0.049   |  0.055    |   0.141  |   0.577 |
> |LLMob-L (Llama 3-8B)      |   0.054   |    0.063  |  0.119   |  0.566  |
> | LLMob-E (GPT-3.5-turbo)      |   0.053   |  0.046    |   0.125  |   0.559 |
> | LLMob-E (GPT-4o-mini)      |   0.041   |   0.053    |   0.211   |   0.531 |
> | LLMob-E (Llama 3-8B)      |  0.054   |   0.059   |  0.122    |   0.561  |
>
> We observed competitive performance of our framework when other LLMs were used. In particular, Llama 3-8B is overall the best when spatial and temporal factors are evaluated together (DARD and STVD). Such results demonstrate the robustness of our framework across different LLMs. On the downside, there are a few cases when Llama 3-8B generated locations that do not exist. Such hallucination was not observed on GPT-3.5-turbo and GPT-4o-mini. Nonetheless, we expect that the promising results of this open LLM can inspire further studies, particularly in developing better fine-tuning techniques for human mobility modeling.

---

> > ### Comment · Reviewer_c5UM · 2024-08-09
> >
> > I would like to thank the authors for their detailed rebuttal. However, I will be maintaining my original score.

---

> > > ### Author Response · Authors · 2024-08-09
> > > **Re: Official Comment by Reviewer c5UM**
> > >
> > > Thank you very much for kindly taking the time to respond to our rebuttal. We greatly appreciate your valuable comments and suggestions. They will be reflected in the revised paper.

---

### Official Review · Reviewer_7xtG · 2024-06-29

**Soundness:** 3
**Presentation:** 2
**Contribution:** 3
**Rating:** 5
**Confidence:** 3

**Summary:**

This paper proposes an LLM-based agent framework for generating personal mobility. It tries to address several problems in the domain of personal mobility generation, including aligning LLMs with the urban data in the real world, developing strategies with reliable activities, and exploring further applications in this field. For generating activities of personal mobility (i.e., trajectories), it proposes a framework of LLM-based agent with an action module, memory module, and planning module, which of all these modules can improve the alignment of the results of generation and real-world data. The paper provides several experiments to verify the effectiveness and alignment of the proposed methods and explores further enhancement.

**Strengths:**

1. This paper is the first simulation with LLM-based agents to generate personal mobility, which can contribute to many fields, such as recommender systems and human behavior studies.
2. The proposed pipeline is intuitive, where the action, memory, and planning modules are important for improving the simulation alignment.
3. The motivation is practical, and the experiments are abundant as well.

**Weaknesses:**

1. What about other LLMs performing as the core, besides GPT-3.5-turbo? I think GPT-4/GPT-4o can be utilized to generate data of higher quality.
2. For the method of Learning-based Motivation Retrieval, I'm a bit concerned about the calculation of similarity among dates. Although the locations are greatly important to identify, there seems to be too much information lost from their initial motivations. Could you try just using cosine similarities on the semantic embeddings of contents with language models? Or maybe using LLMs with prompting to get the score of similarity?
3. I think the presentation can be improved. A preliminary can be added, and the overview of LLMob can be more clear at the start of section 3.

**Questions:**

Please see the above "Weakness". I'm willing to improve my rating if the authors can address my concerns.

**Limitations:**

The authors have discussed the limitations in the last section.

---

> ### Author Rebuttal · Authors · 2024-08-05
>
> Thank you for your insightful comments. We would like to address your concerns as follows.
>
> > **Weakness 1** *What about other LLMs performing as the core, besides GPT-3.5-turbo? I think GPT-4/GPT-4o can be utilized to generate data of higher quality.*
>
> **Answer**: We conducted experiments using GPT-4o-mini and Llama 3-8B. The results are reported as follows:
>
> | Model          | SD | SI | DARD | STVD |
> |----------------|----------|-----------|--------|--|
> | LLMob-L (GPT-3.5-turbo)      |   0.049   |  0.054    |   0.126  |   0.570 |
> | LLMob-L (GPT-4o-mini)      |   0.049   |  0.055    |   0.141  |   0.577 |
> |LLMob-L (Llama 3-8B)      |   0.054   |    0.063  |  0.119   |  0.566  |
> | LLMob-E (GPT-3.5-turbo)      |   0.053   |  0.046    |   0.125  |   0.559 |
> | LLMob-E (GPT-4o-mini)      |   0.041   |   0.053    |   0.211   |   0.531 |
> | LLMob-E (Llama 3-8B)      |  0.054   |   0.059   |  0.122    |   0.561  |
>
> We observe competitive performance of our framework when other LLMs are used. In particular, GPT-4o-mini is the best in terms of the spatial metric (SD); GPT-3.5-turbo is the best in terms of the temporal metric (DI). Llama 3-8B is overall the best when spatial and temporal factors are evaluated together (DARD and STVD). Such results demonstrate the robustness of our framework across different LLMs.
>
> > **Weakness 2** *For the method of Learning-based Motivation Retrieval, I'm a bit concerned about the calculation of similarity among dates. Although the locations are greatly important to identify, there seems to be too much information lost from their initial motivations. Could you try just using cosine similarities on the semantic embeddings of contents with language models? Or maybe using LLMs with prompting to get the score of similarity?*
>
> **Answer**: We agree with your comment and we indeed considered such scheme when doing this work.
>
> We used the open-source sentence embedding model (sentence-transformers/paraphrase-distilroberta-base-v1) to embed the daily activities into a 768-dimensional vector, and used cosine similarity to measure the similarity. We evaluated this setting in the same Tokyo dataset as in the paper. The results are reported as follows:
>
> | Situation          | SD | SI | DARD | STVD |
> |----------------|----------|-----------|--------|--|
> | Normal Trajectory, Normal Data |    0.047  |    0.046   |    0.132 |  0.560   |
> | Abormal Trajectory, Abormal Data |  0.058  |     0.052  | 0.148      | 0.581  |
> | Abormal Trajectory, Normal Data |    0.061   |   0.050   |   0.133   |   0.551  |
>
> We find that when we conduct learning-based motivation retrieval based on sentence embedding similarity, the performance is competitive with the original one. This result demonstrates that the learning-based motivation retrieval scheme is effective in personal mobility generation and that our framework is robust against varying similarity metrics. Considering there are various sentence embedding models and metrics,  the learning-based motivation retrieval scheme based on the similarity of temporal information (i.e., date) is one of the key contributions of our framework.
>
> > **Weakness 3** *I think the presentation can be improved. A preliminary can be added, and the overview of LLMob can be more clear at the start of section 3.*
>
> **Answer**: We appreciate your feedback on improving the paper's presentation. We agree that adding a preliminary section would be beneficial to establish key concepts and definitions before delving into our method. We will also enhance the overview of LLMob at the start of Section 3.

---

> > ### Comment · Reviewer_7xtG · 2024-08-08
> >
> > Thanks for the rebuttal by the authors. I would like to raise my score to 5.

---

> > > ### Author Response · Authors · 2024-08-08
> > > **Response to Reviewer's Engagement**
> > >
> > > Thank you very much for kindly taking the time to respond to our rebuttal! We also greatly appreciate your valuable comments and suggestions. They will be reflected in the revised paper.

---

### Official Review · Reviewer_xLxx · 2024-07-01

**Soundness:** 2
**Presentation:** 2
**Contribution:** 2
**Rating:** 4
**Confidence:** 4

**Summary:**

This paper introduces LLMob, a framework for personal mobility generation using a large language model. The framework aims to leverage urban activity patterns for emulating urban residents, facilitating the human mobility trajectory generation. Using the Tokyo personal activity dataset, the effectiveness of the proposed framework are validated.

**Strengths:**

- The authors proposed LLMob, an interesting human mobility generation framework with Large Language Models.
- The authors provide an illustrative analysis of how LLM can generate reliable activity strategies.

**Weaknesses:**

- The proposed framework has only been validated in GPT-3.5. It remains unclear whether the performance would vary with different backbone models, such as Llama-2.
- The constructed Tokyo dataset statistic is not very clear in this paper, which makes it difficult for the reader to evaluate the true contribution of this paper.
- The difference between this work and existing work (such as [1]) is not well-discussed in the related work section, although the author claims why these methods cannot be easily adopted as the baselines.

[1] Shao, Chenyang, et al. "Beyond Imitation: Generating Human Mobility from Context-aware Reasoning with Large Language Models." arXiv preprint arXiv:2402.09836 (2024).

**Questions:**

- In Line 250 - Line 253, 100 users are chosen and 10 candidate personas are used for subsequent pattern generation. Why did the author select such a setting, I don't see a clear illustration of these settings in the paper.
- In Line 46 and Line 45, the author mentioned "unseen tasks" and "unseen scenarios". In my perspective, the personal mobility generation task is well-formulated in this paper, what does "unseen" mean here for those data-driven methods?

**Limitations:**

In this paper, all experiments are conducted on a single collected dataset (perhaps thousands of trajectories) from Tokyo, which makes this paper not comprehensive enough. Extending the proposed method to more cities helps improve paper quality.

---

> ### Author Rebuttal · Authors · 2024-08-01
>
> Thank you for your insightful comments and time. We would like to address your concerns as follows.
>
> > **Weakness 1** *The proposed framework has only been validated in GPT-3.5. It remains unclear whether the performance would vary with different backbone models, such as Llama-2.*
>
> **Answer**: We conducted experiments using other LLMs (GPT-4o-mini and Llama 3-8B). The results are reported as follows:
>
> | Model          | SD | SI | DARD | STVD |
> |----------------|----------|-----------|--------|--|
> | LLMob-L (GPT-3.5-turbo)      |   0.049   |  0.054    |   0.126  |   0.570 |
> | LLMob-L (GPT-4o-mini)      |   0.049   |  0.055    |   0.141  |   0.577 |
> |LLMob-L (Llama 3-8B)      |   0.054   |    0.063  |  0.119   |  0.566  |
> | LLMob-E (GPT-3.5-turbo)      |   0.053   |  0.046    |   0.125  |   0.559 |
> | LLMob-E (GPT-4o-mini)      |   0.041   |   0.053    |   0.211   |   0.531 |
> | LLMob-E (Llama 3-8B)      |  0.054   |   0.059   |  0.122    |   0.561  |
>
> We observe competitive performance of our framework when other LLMs are used. In particular, GPT-4o-mini is the best in terms of the spatial metric (SD); GPT-3.5-turbo is the best in terms of the temporal metric (DI). Llama 3-8B is overall the best when spatial and temporal factors are evaluated together (DARD and STVD). Such results demonstrate the robustness of our framework across different LLMs.
>
> > **Weakness 2** *The constructed Tokyo dataset statistic is not very clear in this paper, which makes it difficult for the reader to evaluate the true contribution of this paper.*
>
> **Answer**: We agree that detailed data information is important. We have attached a file (please see the pdf file in the above "more detailed information about the experimental data") for more statistics about the data (also including another dataset of Osaka, as requested by the reviewer). We will include the above statistics into Appendix C.2 in the next version of the paper.
>
> > **Weakness 3** *The difference between this work and existing work (such as [1]) is not well-discussed in the related work section, although the author claims why these methods cannot be easily adopted as the baselines.*
>
> *[1] Shao, Chenyang, et al. "Beyond Imitation: Generating Human Mobility from Context-aware Reasoning with Large Language Models." arXiv preprint arXiv:2402.09836 (2024).*
>
> **Answer**: We acknowledge that the related work section could benefit from a more detailed discussion of the differences between our approach and existing methods. For the related work mentioned by the reviewer, there are two major differences from our work:
>
> 1. Our framework is a data-driven framework to exploit the intelligence of LLM. All the components in our framework employs the LLM, except the learning-based motivation retrieval, which employs contrastive learning. In contrast, in [1], an analytical model is adopted to derive personal activities.
>
> 2. Our framework features a consistency evaluation scheme (Section 3.1.2) to align LLMs with the personal activity trajectory data. We did not find such consistency alignment component in [1].
>
> > **Question 1** *In Line 250 - Line 253, 100 users are chosen and 10 candidate personas are used for subsequent pattern generation. Why did the author select such a setting, I don't see a clear illustration of these settings in the paper.*
>
> **Answer**: We sample 100 users to balance between diversity and token consumption. The candidate personas are drafted by first asking the LLM to give us a pool of candidate personas and then we manually select a subset for representativeness. The candidate personas can be regarded as prior knowledge during the generation. They are reported in Table 4 (Appendix D.1).
>
> > **Question 2** *In Line 46 and Line 45, the author mentioned "unseen tasks" and "unseen scenarios". In my perspective, the personal mobility generation task is well-formulated in this paper, what does "unseen" mean here for those data-driven methods?*
>
> **Answer**: By "unseen tasks" and "unseen scenarios", we meant that LLMs generalize to the tasks and scenarios which are not in its training data [1].
>
> [1] Sanh et al. Multitask Prompted Training Enables Zero-Shot Task Generalization. ICLR 2022.
>
> By prompting the LLM to behave like a citizen, we expect that the model can generate activities in certain scenarios that may not have appeared in the data. For example, given the data during pre-pandemic period, the "unseen scenarios" are referred to the situations during the pandemic.
>
> > **Limitations** *In this paper, all experiments are conducted on a single collected dataset (perhaps thousands of trajectories) from Tokyo, which makes this paper not comprehensive enough. Extending the proposed method to more cities helps improve paper quality.*
>
> **Answer**: We conducted an experiment based on the data collected in Osaka, Japan. We generated 537 trajectories based on the 2102 daily activity trajectories from 30 persons. The results are reported as follows, where LLMob-L/E are ours and DiffTraj and TrajGAIL are the best-performing baseline methods.
>
> | Model          | SD | SI | DARD | STVD |
> |----------------|----------|-----------|--------|--|
> | LLMob-L       | 0.035    | 0.021      | 0.141   | 0.391     |
> | LLMob-E        |    0.030  |   0.018    |  0.121   |   0.380    |
> | DiffTraj        |   0.080    |   0.177   |    0.406 |    0.691   |
> | TrajGAIL        |    0.281   |  0.063    |  0.525   |    0.483 |
>
> The above results demonstrate that our framework can maintain superior performance in another city. In Section 5 - Limitations, we acknowledged that it is challenging to collect sufficient data from different areas, which limits our ability to conduct more extensive experiments. Additionally, we note that the model's generalization ability can be also demonstrated by its performance across different scenarios in our original experiments, such as under normal periods and under pandemic periods.

---

> > ### Comment · Reviewer_xLxx · 2024-08-08
> > **Response to The Author**
> >
> > Thanks for the responses, which partially demonstrate the effectiveness of the proposed method on other LLM backbones and datasets. However, the author fails to answer if previous methods cannot be easily adopted as the baselines (mentioned in weakness 3), and fails to provide more well-illustrative reasons for several settings in this paper (mentioned in questions 1 and question 2, the author just explains this again, but didn't provide a reasonable explanation). Considering this paper also has limitations on experimental settings, which still need time and effort to refine, I will keep my score, and I hope these comments could help the author further improve their paper quality.
> >
> > Best,

---

> ### Author Response · Authors · 2024-08-08
> **Response to new comments from reviewer xLxx**
>
> Thanks for acknowledging our response and comments. We would like to highlight our arguments:
>
> 1. Regarding Weakness 3, from the review comments, we did not find the reviewer requiring an explanation on whether previous methods can be easily adopted as baselines. In addition, for the paper mentioned by the reviewer, its source code and prompt were **not available** (the repo was expired when we accessed it, and the paper did not mention what LLM was used) at the time of our submission to NeurIPS.
>
> 2. We are sorry that the reviewer is still confused with Question 1.
>
> 3. For Question 2, our claim in the introduction is not about the experimental setting. It is essentially a claim on the advantages of using LLMs in real-world applications.

---

> ### Comment · Reviewer_xLxx · 2024-08-08
> **Response to The Author**
>
> - **About weakness 3: the difference between this work and existing work**
>
> Although the authors claim the difference in their framework (e.g., it is a data-driven framework and features a consistency evaluation scheme), they didn't explain the advantage behind this. The difference means for what aspect makes this work different from existing work and what its potential contribution is, instead of just explain the difference in module-desgin. Unfortunately, this suggestion appears to have been disregarded by the authors.
>
> - **About question 1**
>
> What I want to know is the motivation or reason behind this setting, e.g., if 100 users and 10 candidates is sufficiently representative of this problem. The author should present this but they didn't.
>
> - **About question 2: "unseen" mean here for those data-driven methods?**
>
> What I want to know is the definition of "unseen" for previous data-driven methods. It seems that the author explains the "unseen" for LLMs and ignores this. Is there a possibility that several traditional methods can deal with "unseen" tasks? Overall, I think the current explanation is not well-illustrative.
>
> - **About limitations on experimental setting**
>
> We the authors demonstrate the proposed framework on Osaka, but the validation part is still not comprehensive enough due to limited sample sizes (only thousands of trajectories here in Tokyo). Regarding the supplementary experiments on GPT-3.5,  GPT-4, and Llama 3-8B, the author seems to not explain which setting they use. Is it conducted on the (Normal Trajectory, Normal Data) setting?
>
> - **Other weakness**
> 1. The latency and the cost of invoking gpt3.5 API is not reported in this paper, which makes the efficiency of such an LLM-based framework remain a potential issue.
> 2. The details of dataset construction (section 4.1) are still not clear. For an LLM-based application, it is very important to clearly present these details, and this will make it easy for other researchers to follow.

---

> > ### Author Response · Authors · 2024-08-08
> > **Response to new comments from reviewer xLxx (1/2)**
> >
> > Thanks for the constructive discussion. We would like to respond as follows:
> >
> > > **Weakness 3**: *Although the authors claim the difference in their framework (e.g., it is a data-driven framework and features a consistency evaluation scheme), they didn't explain the advantage behind this. The difference means for what aspect makes this work different from existing work and what its potential contribution is, instead of just explain the difference in module-desgin. Unfortunately, this suggestion appears to have been disregarded by the authors.*
> >
> > In the previous response, we summarized two significant differences compared to the work mentioned by the reviewer (MobiGeaR): **data-driven** framework and **data alignment** scheme.
> >
> > 1. A data-driven model has the following advantage: it can continuously learn and improve from new data. As more data becomes available, the model's performance can be enhanced, making it adaptive to evolving patterns and trends. Specifically, we demonstrated that our model can automatically adjust to changes in mobility patterns caused by events like the pandemic. In contrast, the mechanistic gravity model used in MobiGeaR is an analytical model, which does not have the capability of learning from and adapting to data. This difference becomes crucial in real-world applications where urban mobility patterns are constantly changing due to factors like urban development, policy changes, or societal shifts.
> >
> > 2. The data alignment scheme has the advantage of ensuring the self-consistency of the identified pattern. The importance of ensuring self-consistency in using LLMs has been investigated in previous studies such as [1]. We did not find such a self-consistency mechanism in MobiGeaR. This feature is of importance for the fidelity of simulated trajectories to actual human behavior is paramount.
> >
> > [1] Wang X, Wei J, Schuurmans D, et al. Self-consistency improves chain of thought reasoning in language models[J]. arXiv preprint arXiv:2203.11171, 2022.
> >
> > To reflect the reviewer's concern, we would like to include a discussion on the differences from MobiGeaR in the next version of the paper.
> >
> > > **Question 1**: *What I want to know is the motivation or reason behind this setting, e.g., if 100 users and 10 candidates is sufficiently representative of this problem. The author should present this but they didn't.*
> >
> > Increasing the number of users and candidate personas improves the sufficiency and diversity of data. On the downside, this compromises the efficiency. We choose such numbers to balance between the two factors. We found that the choice of 100 users and 10 candidate personas is good enough for promising results.
> >
> > In the next version, we would like to include the above explanation.
> >
> > > **Question 2**: *What I want to know is the definition of "unseen" for previous data-driven methods. It seems that the author explains the "unseen" for LLMs and ignores this. Is there a possibility that several traditional methods can deal with "unseen" tasks? Overall, I think the current explanation is not well-illustrative.*
> >
> > By the meaning of "unseen", we meant that the data or scenarios are unseen to the model.
> >
> > For traditional mobility generation methods, we do not think they can handle unseen tasks, because "unseen" refers to data points or patterns that fall outside the distribution of the training data. These methods often struggle with extrapolation to out-of-distribution scenarios. In contrast, our LLM-based approach leverages semantic understanding and general knowledge to reason about the scenarios on which it has not been trained.
> >
> > In addition, such adaptation to unseen scenarios is easy in LLMs by using only a prompt.
> >
> > > **Limitation**: *We the authors demonstrate the proposed framework on Osaka, but the validation part is still not comprehensive enough due to limited sample sizes (only thousands of trajectories here in Tokyo). Regarding the supplementary experiments on GPT-3.5, GPT-4, and Llama 3-8B, the author seems to not explain which setting they use. Is it conducted on the (Normal Trajectory, Normal Data) setting?*
> >
> > We used the (Normal Trajectory, Normal Data) setting. Due to the limited time during the rebuttal period, we cannot cover all the settings. Nonetheless, we would like to report a comprehensive evaluation of the Osaka dataset in the next version of the paper.

---

> > ### Author Response · Authors · 2024-08-08
> > **Response to new comments from reviewer xLxx (2/2)**
> >
> > > **Other Weakness 1**: *The latency of the cost of invoking gpt3.5 API is not reported in this paper, which makes the efficiency of such an LLM-based framework remain a potential issue.*
> >
> > For GPT-3.5 API, the average generation time per trajectory for the results reported in Table 1 is 46 seconds.
> >
> > We would like to report this in the next version of the paper. Moreover, if users seek better efficiency, we suggest using open models such as Llama 3-8B, seeing its promising results as well.
> >
> > > **Other Weakness 2**: *The details of dataset construction (section 4.1) are still not clear. For an LLM-based application, it is very important to clearly present these details, and this will make it easy for other researchers to follow.*
> >
> > For researchers who are interested in our work, we will provide such details in the open-source repo.

---

> > > ### Comment · Reviewer_xLxx · 2024-08-12
> > > **Thanks**
> > >
> > > After carefully reading the author's response, partial concerns were addressed. However, some important parts are still not clear: (1) The dataset construction details, which are very important for the researcher to follow this work; (2) the Lack of statistically rigorous reason behind some settings, e.g., if 100 users and 10 candidates ); (3) The validation size is the obvious limitation of this paper.
> > >
> > > Hope these comments and suggestions could help the author further improve their paper quality.
> > >
> > >
> > > Best,

---

> ### Author Response · Authors · 2024-08-12
> **Thank you for the comments**
>
> Thank you for your time and the valuable comments on this paper. We would like to address your concerns as much as possible:
>
> > **Concern 1**:  *The dataset construction details, which are very important for the researcher to follow this work.*
>
> **Answer:** We agree with the necessity of explaining the dataset construction. In this paper, the data format has been given in Table 2, and dataset statistics has been reported in the attached pdf file in the above "more detailed information about the experimental data".
>
> We obtained the dataset through Twitter (now X)'s Academic Research Product Track. The construction of the dataset is reported as follows:
>
> 1. **Filtering Incomplete Data**: Users with missing check-ins for a specific year were filtered out.
>
> 2. **Excluding Non-Japan Check-ins**: Check-ins that occurred outside of Japan were removed.
>
> 3. **Inferring Prefecture from GPS Coordinates**: Prefectures were inferred based on the latitude and longitude data of check-ins.
>
> 4. **Assigning Prefecture**: Users were assigned to a prefecture based on their primary check-in location; e.g., users whose top check-in location is Tokyo were categorized as belonging to Tokyo.
>
> 5. **Removing Sudden-Move Check-ins**: Check-ins showing abrupt, unrealistic location changes, such as from Tokyo to the Osaka within a very short time frame, were deleted to remove data drift (i.e., fake check-ins), following the criteria proposed by [1].
>
> [1] Yang, D., Zhang, D., & Qu, B. (2016). Participatory cultural mapping based on collective behavior data in location-based social networks. ACM Transactions on Intelligent Systems and Technology (TIST), 7(3), 1-23
>
> 6. **Anonymizing Data**: Real user IDs and geographic location names were anonymized. Only category information of Geographic location was kept, and latitude and longitude coordinates were converted into IDs before being input into the model.
>
> In the next version of the paper, we will cover the construction process in the appendix, including the raw data collection and the preprocessing. We will also provide open data demos and codes related to this research.
>
> > **Concern 2**:  *the Lack of statistically rigorous reason behind some settings, e.g., if 100 users and 10 candidates*
>
> **Answer:** We would like to emphasize that the chosen settings, such as 100 users and 10 candidates, were selected based on practical considerations and the specific context of our study. These settings were chosen to manage a balance between computational efficiency and the ability to demonstrate the effectiveness of our model. The effectiveness has been shown in the experimental section, and the efficiency result has been reported in the above discussion.
>
> > **Concern 3**: *The validation size is the obvious limitation of this paper.*
>
> **Answer:** We agree that having a larger validation size would be beneficial in demonstrating the model's performance. However, due to constraints including the data availability and computational resources, we had to make practical decisions regarding the validation size in this research. We believe that our chosen setting is reasonable, which is partly supported by the similar evaluation setting taken within the 25-LLM-agent community constructed in [1].
>
> [1] Park J S, O'Brien J, Cai C J, et al. Generative agents: Interactive simulacra of human behavior. Proceedings of the 36th annual ACM symposium on user interface software and technology. 2023: 1-22.

---

> ### Author Response · Authors · 2024-08-12
> **Re: Thanks**
>
> Following our previous post, in which we tried to address the most recent concerns of the reviewer, we sincerely appreciate the reviewer's dedicated efforts in evaluating our paper and engaging in the discussion. Throughout the discussions, we have tried our best to address the raised questions, particularly regarding the dataset details (construction & statistics) and the inclusion of additional experiments. We find these discussions highly constructive for improving the quality of our paper.
>
> Regarding the identified limitations, we acknowledge that expanding the number of users and candidate personas could potentially enhance the data's sufficiency and diversity. However, we wish to emphasize that our current selection of 100 users and 10 personas has already yielded **promising outcomes**, as demonstrated in our results. It is important to note that increasing these numbers would lead to **higher token consumption and computational costs**. This limitation is not unique to our study but is also **observed in many notable existing works** that utilize GPT 3.5/4 APIs, including the seminal work of generative agents [1], which involved only 25 agents.
>
> [1] Park J S, O'Brien J, Cai C J, et al. Generative agents: Interactive simulacra of human behavior. Proceedings of the 36th annual ACM symposium on user interface software and technology. 2023: 1-22.
>
> We are committed to addressing all concerns highlighted by the reviewer in the next version of our paper, incorporating all details from this discussion. We hope that these efforts meet the reviewer's expectations and merit consideration for a higher rating.

---

### Official Review · Reviewer_abCw · 2024-07-11

**Soundness:** 3
**Presentation:** 2
**Contribution:** 2
**Rating:** 6
**Confidence:** 4

**Summary:**

This paper presents a prompt engineering framework for generating synthetic human trajectories using LLMs. The framework is guided, in an overall manner, by the observation that human movement is affected by habitual activity patterns and motivations.  In this way, the framework has two phases for considering these aspects. In phase 2, two varying strategies are proposed and compared. All in all, this is an interesting piece of application work of LLMs, in prompt engineering.

**Strengths:**

1)	A general solid prompt engineering framework for an interesting application in cities.
2)	The method is non-trivial and has a certain level of novelty.
3)	The evaluation has been comprehensive.

**Weaknesses:**

1)	There are some parts in the text that are difficult to decipher – I wouldn’t say the paper is easy to follow.
2)	It is unclear which method actually yields the best performance across different metrics
3)	The generalizability is unclear. Can this method be used in other cities around the globe?

**Questions:**

1)	Is this method zero-shot?
2)	How is the Td in 3.2 (habitual activity pattern) used subsequently?
3)	What’s the relation between 3.2.1 evolving based motivation retrieval and 3.2.2 learning-based motivation retrieval?
4)	Learning-based motivation retrieval appears to be dependent on embeddings obtained through a contrastive learning process. In what way the embeddings are actually used?
5)	What’s the rationale behind the two retrieval strategies? If one is superior, why the other is being discussed?

**Limitations:**

See weaknesses

---

> ### Author Rebuttal · Authors · 2024-08-01
>
> Thank you for your insightful review and valuable feedback on our work. We would like to address the concerns as follows.
>
> > **Weakness 2**: *It is unclear which method actually yields the best performance across different metrics.*
>
> **Answer**: In our evaluation, we employ four metrics to comprehensively assess mobility generation: SD is a spatial metric, SI is a temporal metric, and DARD and STVD evaluate both spatial and temporal factors.
>
> Although there is no method that outperforms all the others in these metrics, the proposed framework delivers an overall best performance across these metrics. In particular, it achieves the best in DARD and SI, and the runner-up in STVD. That is, our framework excels in reproducing temporal and spatio-temporal aspects of personal mobility. Such result has also been shown effective in a real-world application as event-driven generation (Section 4.2).
>
> > **Weakness 3**: *The generalizability is unclear. Can this method be used in other cities around the globe?*
>
> **Answer**: The proposed framework is inherently area-agnostic. The core idea of our framework is based on personal pattern identification and motivation retrieval, both of which are data-driven. This ensures that the method can be applied universally given sufficient data, as it leverages locally available check-in data to derive activity patterns.
>
> To further validate our claim, we conducted an experiment based on the data collected in Osaka, Japan. We generated 537 trajectories based on the 2102 daily activity trajectories from 30 persons. The results are reported as follows, where LLMob-L/E are ours and DiffTraj and TrajGAIL are the best-performing baseline methods.
>
> | Model          | SD        | SI        | DARD      | STVD      |
> |----------------|-----------|-----------|-----------|-----------|
> | LLMob-L        | 0.035     | 0.021     | 0.141     | 0.391     |
> | LLMob-E        | 0.030     | 0.018     | 0.121     | 0.380     |
> | DiffTraj       | 0.080     | 0.177     | 0.406     | 0.691     |
> | TrajGAIL       | 0.281     | 0.063     | 0.525     | 0.483     |
>
> The above results demonstrate that our framework can maintain superior performance in another city. In Section 5 - Limitations, we acknowledged that it is challenging to collect sufficient data from different areas, which limits our ability to conduct more extensive experiments. Additionally, we note that the model's generalization ability can be also demonstrated by its performance across different scenarios in our original experiments, such as under normal periods and under pandemic periods.
>
> > **Question 1**: *Is this method zero-shot?*
>
> **Answer**: Yes, our method is zero-shot. While we use historical data to extract general activity patterns and create a pool of potential locations, the actual trajectory generation process is zero-shot. The LLM agent does not rely on seeing complete trajectory examples for the specific day or scenario it is generating for. Instead, it uses the extracted patterns and motivation retrieval to reason about and create entirely new, unseen trajectories. This zero-shot capability allows our method to generate plausible activities even for novel scenarios, such as the COVID-19 pandemic example in our experiments, without requiring any direct examples of trajectories in those conditions.
>
> > **Question 2**: *How is the Td in 3.2 (habitual activity pattern) used subsequently?*
>
> **Answer**: The habitual activity pattern derived from equation (2) in 3.2 is used as part of the prompt to retrieve the motivation, i.e., the planning of the daily activity (Please see Appendix B: Page 14, Line 509 <INPUT 0> and Line 513 <INPUT 0>). In this way, we expect the LLM to behave as a citizen following the specified habitual activity pattern.
>
> > **Question 3-5**: *(1) What's the relation between 3.2.1 evolving based motivation retrieval and 3.2.2 learning-based motivation retrieval? (2) Learning-based motivation retrieval appears to be dependent on embeddings obtained through a contrastive learning process. In what way are the embeddings actually used? (3) What's the rationale behind the two retrieval strategies? If one is superior, why is the other being discussed?*
>
> **Answer**: Regarding evolving-based and learning-based motivation, these two components were designed to explore in two directions to identifying the current motivation for daily activity generation, each dealing with different aspects of data availability and sufficiency. We considered them as two promising directions for designing solutions to real-world applications, rather than determining which is superior. The experimental results also show that neither approach always outperforms the other.
>
> - **Evolving-Based Motivation Retrieval**: This method infers the motivation for daily activities based on the activity data from the past few days. It focuses on **short-term temporal patterns and trends**, adapting to recent changes in behavior or context. This method can be a good candidate when **short-term data is available but long-term data is limited**.
>
> - **Learning-Based Motivation Retrieval**: This method leverages a learning-based model to infer motivations by comparing activities from similar dates. The similarity is evaluated through a model trained on historical data. The trained model is then used to evaluate the similarity of the targeted date and the historical dates. This scheme is designed to identify motivations from a broader temporal context, providing robust insights even when **short-term data is limited but long-term data is abundant**.

---

> > ### Comment · Reviewer_abCw · 2024-08-11
> > **Thanks**
> >
> > I would like to thank the authors for the response, and I would like to raise my score.

---

> > > ### Author Response · Authors · 2024-08-11
> > > **Re: Thanks**
> > >
> > > Thank you very much for kindly taking the time to respond to our rebuttal! We also greatly appreciate your valuable comments and suggestions. They will be reflected in the revised paper.

---

### Author Rebuttal · Authors · 2024-08-07

Thank you to all the reviewers for your comments. We have addressed each reviewer's feedback in the individual response page. Additionally, we have attached a file that provides **more detailed information about the experimental data** we used.

---

### Author Response · Authors · 2024-08-14
**Summary of Reviews and Rebuttals**

**Table Reviewer Rating Summarization.**

| Score/Reviewer | Reviewer abCw | Reviewer xLxx                                         | Reviewer 7xtG | Reviewer c5UM | Average    |
| -------------- | ------------- | ----------------------------------------------------- | ------------- | ------------- | --- |
| Rating         | 6             | 4  | 5             | 6             |  5.25    |
| Confidence    | 4             | 4                                                     | 3             | 4             |  3.75   |

From the table, it is clear that **three reviewers** have been **very positive** towards our paper.

Our work is the **first work** to **employ LLM agent for semantic human mobility generation**, which is a **very important and pioneering attempt in the field of LLM for computational social science**. Hope we could get your credit and kind support. Thanks a lot!

---

### Decision · Program_Chairs · 2024-09-25

**Decision:**

Accept (poster)

**Comment:**

This paper introduces LLMob, a novel approach using Large Language Models (LLMs) integrated into an agent framework for flexible and effective personal mobility generation
The method requires insightful generation of structured prompts. It is brining the idea of agents systems to personal mobility and looks at multiple problems: including aligning LLMs with the urban data in the real world, developing strategies with reliable activities.
The empirical performance is limited, since it is on a biased and limited dataset. However the reviewer acknowledges that availability of such data is not easy, and hence showcasing this new problem may need creation and curation of the data itself. There are a number of concerns, mainly centered on clarity of writing and clarity in terms of empirical superiority of the method. The metrics used are nonintuitive and since there is no single methods that dominates, it is not clear what the value of the presented method is.  However the rebuttal does a decent job of clarifications and additional experiments such as evaluation with Llama.

Furthermore there are ethical concerns on the privacy of the data. The authors confirmed that the data is obtained through publicly available APIs from Foursquare and Twitter, and hence follow their guidelines. The authors should follow through with the plan of additional steps to anonymize the output location depending on the specific requirements of the application.  Given the over all contributions, the meta-reviewers finds that the paper has value of the subcommunity of ML studying mobility with LLMs. The authors are however requested to make major changes improvements to the manuscript based on the discussion feedback to improve writing, include new experiments, acknowledge limitations and suggest solutions for the data anonymity.